# Binding of transmissible gastroenteritis virus and porcine respiratory coronavirus to human and porcine aminopeptidase N receptors as an indicator of cross-species transmission

Mykyta Peka [1,2*], Viktor Balatsky[2]

**1** Department of Molecular Biology and Biotechnology, School of Biology, V. N. Karazin Kharkiv National University, Kharkiv, Ukraine, **2** Genetics Laboratory, Institute of Pig Breeding and Agroindustrial Production, National Academy of Agrarian Sciences of Ukraine, Poltava, Ukraine

\* pekapoltava@gmail.com

## Abstract

Coronaviruses have the ability to overcome interspecies barriers and adapt to new hosts, posing significant epidemic risks in cases of zoonotic transmission to humans. A critical factor in this process is the interaction between coronavirus spike proteins and host cell surface receptors, which plays an important role in infection and disease progression. This study focused on two representatives of coronaviruses: transmissible gastroenteritis virus (TGEV) and its mutant, porcine respiratory coronavirus (PRCV), both of which naturally cause disease in pigs. A phylogenetic analysis of previously identified strains of these viruses was performed, and the conservation of receptor-binding domain (RBD) sequences within their spike proteins was evaluated. *In silico* modeling was performed for complexes of the RBDs from 16 virus strains with porcine aminopeptidase N (APN), as well as for putative complexes with the human APN receptor. The binding free energy of these modeled complexes was evaluated, along with the impact of more than 500 theoretical mutations in the RBD. The computational results suggest that the TGEV 133 strain exhibits the highest affinity for both porcine and human receptors, with only two additional mutations required to further enhance this affinity. Molecular dynamics simulations were conducted for porcine and human APN complexes with known TGEV strains (Purdue and 133) as well as a theoretical mutated strain. These simulations reveal differences in the dynamic behavior of complexes with porcine and human receptors and support the hypothesis that mutagenesis at a few key amino acid residues in the RBD could enable TGEV to achieve affinity for human APN comparable to that of its natural host receptor. The findings underscore a theoretical risk of zoonotic transmission of these coronaviruses to humans, emphasizing the importance of further monitoring these pathogens.

**Data availability statement:** All relevant data are within the manuscript and its Supporting Information files.

**Funding:** This study was funded by the National Academy of Agrarian Sciences of Ukraine (grant registration number: 0124U002088). The funder had no role in study design, data collection and analysis, decision to publish, or preparation of the manuscript.

**Competing interests:** The authors have declared that no competing interests exist.

## Introduction

Viruses are the cause of an extremely large number of diseases, impacting a wide range of biological species [1–3]. Moreover, the majority of extant viruses remain unstudied [4]. Coexistence with constantly evolving viruses in the environment requires humans to be actively engaged in studying their biology, monitoring and surveillance, drug development, etc. Viruses that affect vertebrates have negative consequences by destroying animal populations [5], reducing the productivity of livestock species and causing losses to animal husbandry [6], while also creating potential risks for further transmission to humans [7]. Animals serve as reservoirs for numerous zoonotic infections [8–11] that have repeatedly been spread to humans, leading to epidemics [12–14].

The *Coronaviridae* family, in particular, displays remarkable adaptability in crossing interspecies barriers and acquiring specificity for human infection. Within a relatively short evolutionary period, several coronaviruses have made transmissions from animals to humans: HCoV-229E probably from bats via alpacas or dromedaries 200 years ago; HCoV-OC43 from bovine 120 years ago; SARS-CoV and MERS-CoV from bats via palm civets and dromedary camels, respectively, in recent decades. Two other coronaviruses, HCoV-NL63, and HCoV-HKU1 were first clinically identified in 2004–2005 [15–17]. There is evidence indicating the possible primary origin of HCoV-NL63 and HCoV-229E from bat, and HCoV-OC43 and HCoV-HKU1 from rodent viruses accordingly [18]. Finally, the pandemic SARS-CoV-2 emerged in 2019, but its origin is still under investigation [19]. Given such evolutionary dynamics, particularly in recent years, the possibility of other transmissions of coronaviruses from animals to humans cannot be ruled out.

Among the animals that host coronaviruses, pigs can be distinguished. Six clinically significant coronaviruses are currently known [20], including porcine epidemic diarrhea virus (PEDV), transmissible gastroenteritis virus (TGEV), porcine respiratory coronavirus (PRCV) and swine acute diarrhea syndrome coronavirus (SADS-CoV) which belong to *Alphacoronavirus* genus, porcine hemagglutinating encephalomyelitis virus (PHEV) which belongs to *Betacoronavirus* genus, and porcine deltacoronavirus (PDCoV) which belongs to *Deltacoronavirus* genus. Previous report [17] suggest that pigs have the potential to serve as a reservoir for recombination of coronaviruses, posing a clear danger for the emergence and spread of new viruses in the environment. TGEV and PRCV are of particular interest for study, as the latter virus differs from the former in the presence of deletions in the S and ORF3 genes [21]. PRCV is thus considered a TGEV mutant [22]. These deletions cause changes in the virus's pathogenic potential and tropism: TGEV infects intestinal enterocytes, resulting in pig diarrhea, while PRCV leads to respiratory tract infection [22]. A deletion in the S gene, encoding the spike protein (S-protein) of these viruses, appears critical for altering the virus's properties [23]. Importantly, while TGEV causes severe disease with extremely high mortality in piglets, PRCV causes subclinical symptoms.

Despite certain differences, PRCV and TGEV exhibit similar antigenic properties and induce comparable immune responses in pigs. Thus, antibodies generated as a

consequence of PRCV infection provide partial protection against TGEV infection in pigs [24–26] and may enhance the efficacy of TGEV vaccination [27]. However, immunization solely with PRCV does not provide complete protection against TGEV [27]. The circulation of PRCV in pig populations is believed to be a contributing factor to the decrease in TGEV incidence in recent years [20]. A review published in 2021 [20] highlights sporadic outbreaks of TGEV during the period 2015–2020, with no documented cases of PRCV infection within the same timeframe. However, subsequent report [28] revealed the detection of PRCV RNA in five swine samples in the United States, with sequencing of one isolate (USA/ISU20–92330/2020) revealing significant genetic distinctions from traditional PRCV strains. TGEV has also undergone genetic alterations in recent years, potentially via recombination events with PRCV [29], suggesting ongoing evolutionary processes in these viruses.

The S-proteins in TGEV and PRCV serve the function of recognizing the receptor on the host cell surface [30], which is porcine aminopeptidase N (pAPN). A functional region responsible for APN recognition and binding has been identified in the S-protein, which can be considered a receptor-binding domain (RBD) [31]. Similarly, HCoV-229E enters human cells by attaching to human aminopeptidase N (hAPN), but porcine and human viruses use different receptor sites [32]. Additionally, there is evidence of the possible attachment of TGEV to mutated hAPN [33].

Studying TGEV and PRCV is of great interest given the impact of these viruses on pig health. Recently, *Coronaviridae* representatives have shown a significant potential for interspecific transmissions and adaptation to the human hosts, the high point of which was the COVID-19 pandemic. S-protein plays a definitely important role in the evolution of coronaviruses; therefore, this study aims to examine the ability of RBD in various TGEV and PRCV strains to interact with pAPN and hAPN, through computational approaches. Additionally, the theoretical potential of these viruses to change their receptor affinity due to the accumulation of mutations is assessed.

## Materials and methods

Primary S-protein amino acid sequences of various TGEV and PRCV strains (isolates) deposited in the NCBI database [34] as of January 2024 were used in the study. For phylogenetic analysis, sequences belonging to different Identical Protein Groups were selected to exclude completely matching amino acid sequences. A total of 65 TGEV and 13 PRCV S-protein sequences were selected. The list of protein sequences with their lengths and NCBI accession numbers is given in S1 Table. In sequences where individual amino acids were missing, they were determined taking into account the genetic code and possible amino acids in the corresponding positions of the sequences of other strains. This procedure was done to avoid unwarranted divergence of such sequences in phylogenetic analysis.

Multiple alignment of the selected sequences was performed using the MUSCLE algorithm [35] in MEGA11 software [36]. The phylogenetic tree was then built using the Maximum Likelihood method and the JTT matrix-based model [37].

Based on the results of clustering, virus strains were selected, and the primary S-protein sequences of these strains were used for further analysis. Among TGEV strains considered were: Purdue (PUR46-MAD) (NCBI RefSeq: NP_058424.1), 133 (GenBank: AFZ88844.1), AHHF (GenBank: AQT01349.1), FS772/70 (UniProtKB/Swiss-Prot: P18450.2), Miller M6 virulent (GenBank: ABG89301.1), USA/Minnesota138/2006 (GenBank: ASV64295.1), SZ19 (GenBank: UOZ96074.1), 96–1933 (GenBank: AAC96004.1), H16 (GenBank: ACN71196.1), TO14 (GenBank: AAG30227.1); and among PRCV strains considered were: HOL87 (GenBank: AAA46905.1), 135 (GenBank: URY50789.1), 137 (GenBank: URY50805.1), AR310 (GenBank: URY50797.1), ISU-1 (GenBank: ABG89317.1), OH7269 (GenBank: AKV62755.1). These sequences were also subjected to a separate multiple alignment in MEGA11, after which a distance matrix was constructed using the Poisson correction model [38]. An extended multiple alignment aimed at assessing evolutionary conservation was also performed, including several S-protein sequences from other coronaviruses: canine coronavirus (CCoV, GenBank: AAV65515.1), PDCoV (GenBank: AKC54428.1), HCoV-229E (NCBI RefSeq: NP_073551.1), SARS-CoV (UniProtKB/Swiss-Prot: P59594.1), SARS-CoV-2 (GenBank: QHD43416.1), and MERS-CoV (NCBI RefSeq: YP_009047204.1).

In positions of RBDs of the S-proteins, where different strains of TGEV and PRCV had amino acid substitutions relative to each other, a pairwise assessment of the conservation of such substitutions was carried out using Grantham's distances [39]. Based on these estimates, amino acid substitutions in RBD between strains were characterized in terms of their conservation according to the existing classification [40].

Homology modeling of three-dimensional RBD-APN complexes was performed using the crystal structure of the spike receptor binding domain of a porcine respiratory coronavirus in complex with the pig aminopeptidase N ectodomain [30] deposited in the RCSB PDB (PDB ID: 4F5C) as a template. For this, sequence fragments of the pAPN (NCBI RefSeq: NP_999442.1), hAPN (NCBI RefSeq: NP_001141.2) and S-protein of Purdue (PUR46-MAD) strain corresponding to the template were aligned with the A and B chains of the template. Further homology modeling was carried out using the SWISS-MODEL Server [41] for RBD of Purdue strain complexed with pAPN and hAPN.

Once the models for the RBD of the Purdue strain with hAPN and pAPN were determined, similar models of complexes involving RBDs of other TGEV and PRCV strains were constructed. For this purpose, corresponding amino acids of the RBDs were mutated using Modeller 10.4 [42].

The binding free energy (ΔG) calculations for all RBD-APN complexes were performed through molecular dynamics simulations on the HawkDock server v1 [43] to evaluate the stability of virus-receptor interactions. The procedure involved minimizing all structures and following calculating ΔG using the Molecular Mechanics/Generalized Born Surface Area (MM/GBSA) method [44].

To predict a possible further increase in the affinity of those virus strains that form stable RBD-APN complexes, the effect of possible mutations in the RBD on changes in binding free energy (ΔΔG) was assessed. A systematic analysis of different mutations was first carried out using BeAtMuSiC [45] and mCSM-PPI2 [46] services. The possibility of mutations occurrence taking into account the genetic code in the corresponding triplets was assessed [47]. Next, the most potentially impacting and possible mutations were simultaneously assessed using BindProfX [48], MutaBind2 [49], and SAAMBE-3D [50] services. Subsequently, mutations were introduced into the RBD chains of the corresponding RBD-APN complexes using the procedure described above, and the stability of the resulting complexes was again assessed by calculating ΔG using the MM/GBSA method on the HawkDock server.

Molecular dynamics of RBD-APN complexes created both for RBDs of existing TGEV and PRCV strains, and for RBDs with potential mutations, was carried out using the GROMACS 2023.3 program [51–53]. The simulations at this stage were carried out using modified CHARMM36 force field [54]. Structures were minimized in an aqueous environment with 150 mM ions, then equilibrated as follows: 1 ns NVT with V-rescale thermostat (temperature coupling constant 0.1 ps), 1 ns NPT with V-rescale thermostat (0.1 ps) and C-rescale barostat (pressure coupling constant 2.0 ps), and 50 ns NPT with V-rescale thermostat (1.0 ps) and C-rescale barostat (5.0 ps). Afterwards, molecular dynamics simulations for 400 ns were performed with V-rescale thermostat (1.0 ps) and C-rescale barostat (5.0 ps). The effects of periodic boundary conditions (PBC) were eliminated, followed by the assessment of root mean square deviation (RMSD), root mean square fluctuation (RMSF), radius of gyration (Rg), solvent accessible surface area (SASA) [55,56], and the number of hydrogen bonds and non-polar contacts.

Afterwards, molecular dynamics trajectories for complexes involving either human or porcine receptors were fitted using backbone atoms, concatenated, and principal component analysis (PCA) was performed on the C-alpha atoms. Based on the projections of the trajectories onto the first two principal components, a free energy landscape (FEL) analysis was conducted for dynamics of each of the RBD-APN models.

To compute ΔG throughout the entire molecular dynamics trajectories, the Molecular Mechanics/Poisson–Boltzmann Surface Area (MM/PBSA) method was employed using the gmx_MMPBSA tool [57,58]. In this case, the ΔG approximated as the enthalpic component without the inclusion of the entropic contribution.

To visualize three-dimensional structures, the open-source PyMOL version 2.5 was used [59], and the evaluation of contacts at the interface of RBD-APN complexes was carried out using PDBsum [60].

## Statistical analysis

Statistical analysis was conducted to compare the acquired ΔG values (on the HawkDock server v1) for complexes of porcine and human APN with different TGEV and PRCV strains' RBDs. Firstly, two groups of complexes, involving pAPN receptors associated with RBDs of either TGEV strains (first group) or PRCV strains (second group), were compared. The distribution of values within each group was assessed for normality using the Shapiro-Wilk test. Additionally, data were examined for outliers using the interquartile range method. Identified outliers were retained in the dataset to preserve the full range of observed variability. The differences between groups were further evaluated using a nonparametric Wilcoxon rank-sum test (Mann–Whitney $U$ test). Differences were considered statistically significant at $p < 0.05$. Subsequently, similar statistical analyses was performed to assess differences between groups of complexes involving hAPN receptors associated with the RBD of either TGEV or PRCV strains. Following this, to assess differences between groups of complexes, encompassing either porcine (first group) or human (second group) receptors with virus strains of the same type, a nonparametric Wilcoxon signed-rank test was used.

In analyzing the molecular dynamics results, statistical methods were used to calculate the mean values and standard deviations for parameters such as Rg, SASA, the number of H-bonds and non-polar contacts, as well as energy parameters throughout the studied molecular dynamics trajectories.

## Results

### Phylogenetic analysis and S-protein sequences comparison

The sequences and lengths of the amino acid chains of the S-protein in different strains of TGEV and PRCV are characterized by variability. For TGEV, S-proteins of 1449 and 1447 amino acid residues predominate, totaling 31 and 22, respectively, from the studied strains. The decrease in length by two amino acid residues is due to the deletion (N375_D376del), which is typical for strains grouped into the appropriate cluster together with the Purdue (PUR46-MAD) strain (Fig 1). Additionally, some strains have an S-protein with other single amino acid deletions: V796del in the absence of N375_D376del results in a chain length of 1448 amino acid residues (Miller M60, attenuated H, H-16, HN-2012, CH/GX/TGEV/2662/2019, DS01–2022, SC2021, CH8438, HLJ-17, JMS, and AHHF strains), and F1408del combined with N375_D376del gives S-protein with a chain length of 1446 amino acid residues (SZ-19 strain).

The S-protein in PRCV differs from that of TGEV by a deletion in the N-terminal region. Although the size of the deletion is variable, it is predominantly 224 amino acid residues with localization C21_G244del, which results in the S-protein size of 1225 amino acid residues (strains 135 (86/135308), 137 (86/137004), 86/137004/British isolate, RM4, 1/90-DK, HOL87). At the same time, other strains may have displaced deletion frames, which may both result in an increase in the chain size in strains OH7269 (1232 aa), USA/ISU20–92330/2020 (1232 aa), USA/AR310/1989_ISU (1242 aa), and its decrease in strains USA/1894X/1992_ISU (1223 aa), AR310 (1222 aa), ISU-1 (1222 aa), USA/Minnesota-46140/2016 (1221 aa). It should also be noted that the size of the S-protein in some strains (USA/Minnesota-46140/2016, USA/ISU20–92330/2020, OH7269) is affected by the Y957del (the position is indicated relative to the length of the TGEV S-protein). On the phylogenetic tree, all PRCV strains are located in two clusters. The first cluster includes all strains with a 224 aa deletion, and the second cluster comprises the other strains (Fig 1).

Taking into account the results of clustering of virus strains by their S-proteins, regional prevalence, and degree of study, 10 strains of TGEV and 6 strains of PRCV were selected as promising subjects for more detailed study. It is important to note that each of these strains has different (non-identical) sequences in the RBD region. The positions at which the S-proteins of these strains have different amino acids within the RBD are indicated on Fig 2 and in S2 Table. The size of the RBD (150 amino acid residues) and its localization were determined based on data from [31]. For TGEV strains without deletions, the RBD is positioned at amino acid residues 524–673, and for strains with N375_D376del, it shifts by

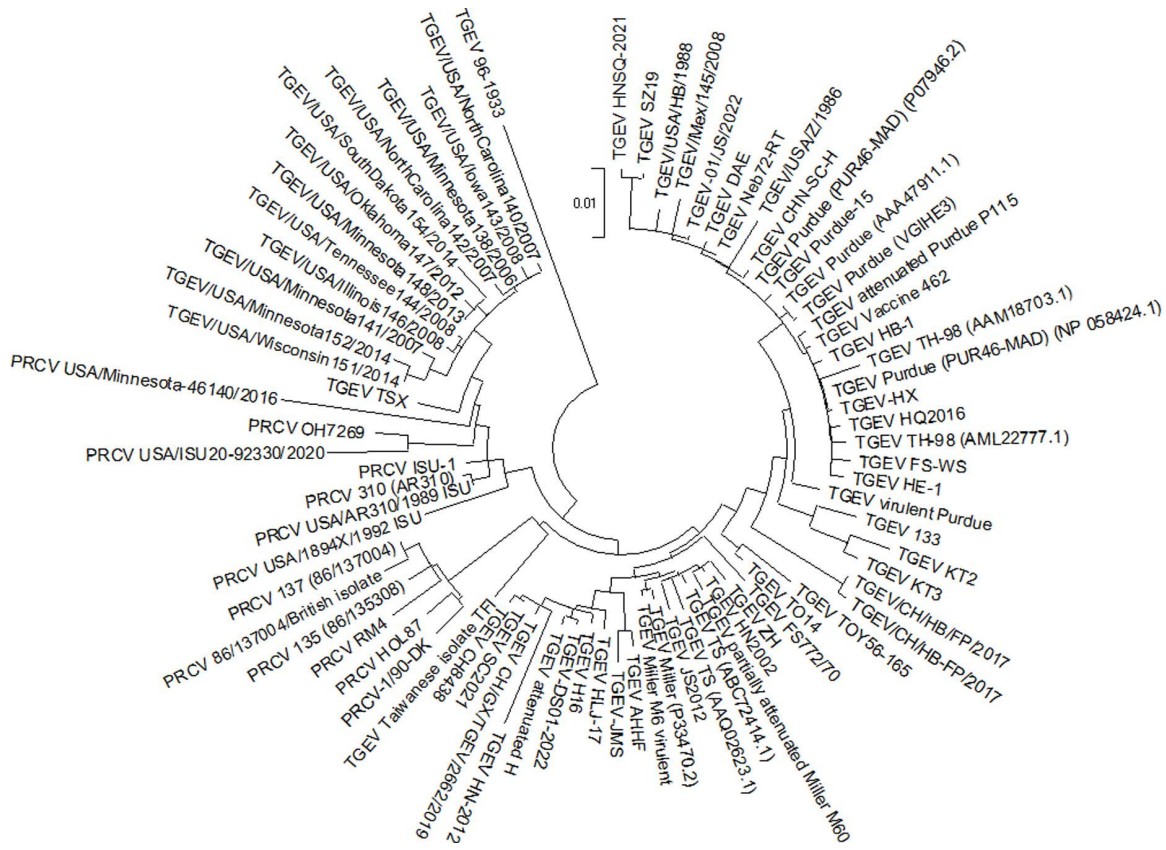

**Fig 1. Phylogenetic tree of TGEV and PRCV strains (isolates) built based on the S-protein amino acid sequences.** The phylogenetic tree was built using the Maximum Likelihood method and the JTT matrix-based model based on the results of multiple alignment of S-protein amino acid sequences performed according to the MUSCLE algorithm.

two amino acids to position 522–671. In PRCV strains with a deletion of 224 amino acids, the RBD is located at positions 300–449; in PRCV strains with a different deletion size, it is shifted by several positions.

Fig 2 presents a multiple sequence alignment of the amino acid sequences of 10 TGEV and 6 PRCV strains at the RBD of the S-protein, along with several other members of the *Coronaviridae* family, including CCoV, HCoV-229E, SARS-CoV, SARS-CoV-2, MERS-CoV, and PDCoV. The amino acid substitutions at the RBD, characteristic of different TGEV and PRCV strains, are color-coded based on Grantham's distances, which characterize the conservatism or radicality of amino acid substitutions (Fig 2). The specific Grantham's distances for these substitutions are provided in S2 Table.

As can be observed, most of the substitutions are of a conservative or moderately conservative nature. In other words, the amino acids in the respective variable positions in different strains predominantly exhibit similar physicochemical properties. When comparing the sequences at the RBD regions of a broader range of coronaviruses, it is evident that CCoV, which belongs to the same *Alphacoronavirus* 1 species as all TGEV and PRCV strains, exhibits a highly similar sequence at the RBD region. Although CCoV contains some substitutions relative to the reference TGEV Purdue strain, several of these substitutions are shared with other TGEV and PRCV strains. Another representative of the *Alphacoronavirus* genus, HCoV-229E, also contains several conserved motifs that are similar in sequence to those of TGEV and PRCV strains.

In contrast, representatives of the *Betacoronavirus* genus (SARS-CoV, SARS-CoV-2, and MERS-CoV) and the *Deltacoronavirus* genus (PDCoV) exhibit significantly different sequences at the aligned regions. This divergence

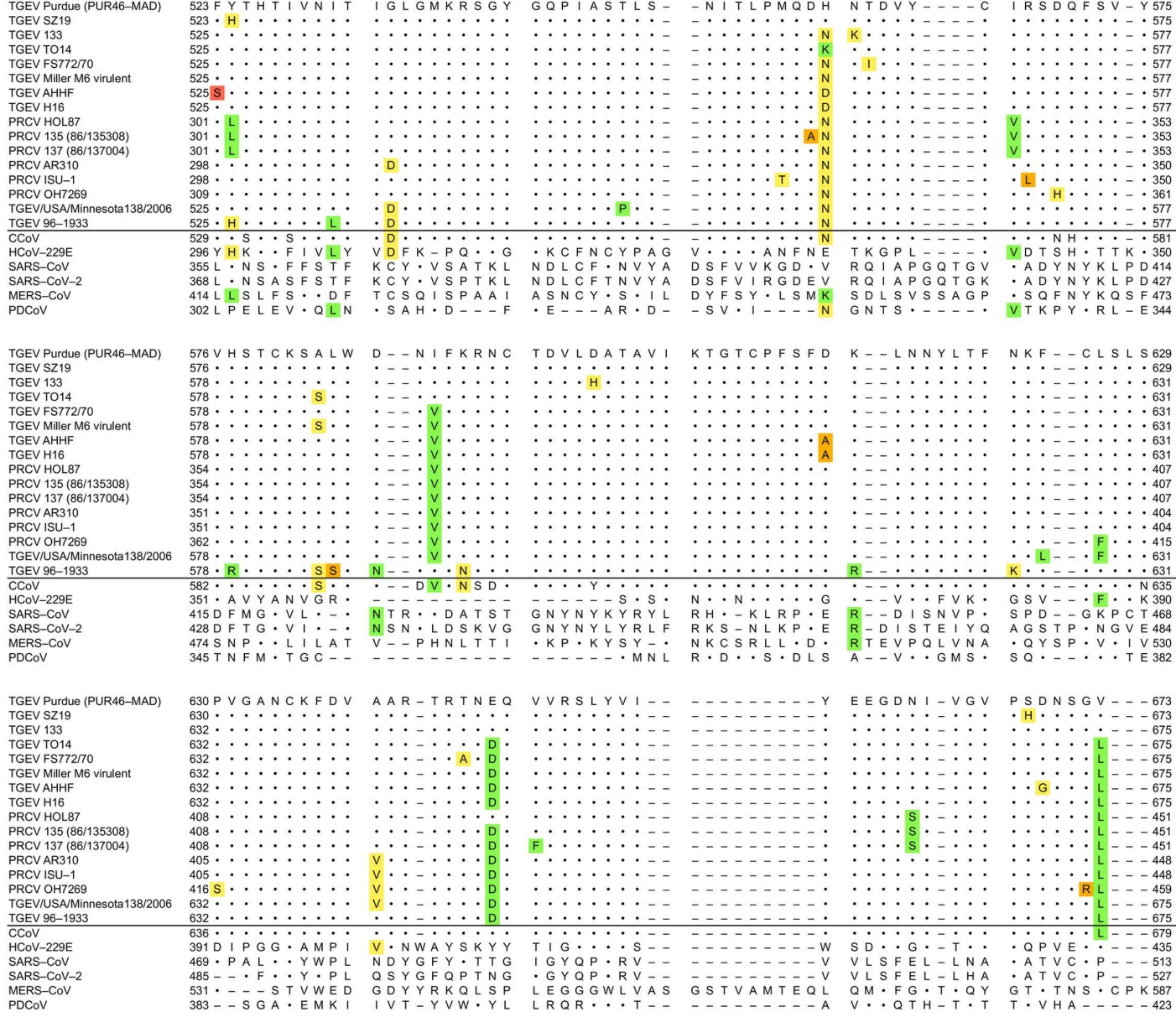

**Fig 2. Comparison of amino acid sequences of the S-protein RBD regions among TGEV, PRCV, and other coronaviruses.** Conservation of amino acid substitutions in TGEV and PRCV strains was assessed using Grantham's distances: conservative (green, 0–50), moderately conservative (yellow, 51–100), moderately radical (orange, 101–150), and radical (red, ≥ 151). Identical substitutions in other coronaviruses were similarly color-coded. The RBD region spans residues 522–671 in the reference TGEV Purdue strain. Residue S522 is omitted due to a gap in multiple alignment with sequences from other coronavirus species; details on substitutions and Grantham's distances are provided in S2 Table.

suggests substantial differences in the RBDs of the S-proteins of these viruses compared to TGEV and PRCV strains.

Table 1 presents a distance matrix built based on S-protein amino acid sequences of the 16 studied strains. The distance matrix, like the phylogenetic tree in Fig 1, illustrates the relationships between strains, categorizing them by indicating pairwise distance values. Consequently, the instances of smallest distances are observed between the strains PRCV

**Table 1. Distance matrix between S-protein amino acid sequences of 16 TGEV and PRCV strains.**

| | | 1 | 2 | 3 | 4 | 5 | 6 | 7 | 8 | 9 | 10 | 11 | 12 | 13 | 14 | 15 |
|---|---|---|---|---|---|---|---|---|---|---|---|---|---|---|---|---|
| 1 | TGEV Purdue (PUR46-MAD) | | | | | | | | | | | | | | | |
| 2 | TGEV SZ19 | 0.01113 | | | | | | | | | | | | | | |
| 3 | TGEV 133 | 0.01392 | 0.02167 | | | | | | | | | | | | | |
| 4 | TGEV TO14 | 0.01532 | 0.02097 | 0.02022 | | | | | | | | | | | | |
| 5 | TGEV FS772/70 | 0.02378 | 0.02876 | 0.02799 | 0.01670 | | | | | | | | | | | |
| 6 | TGEV Miller M6 virulent | 0.01813 | 0.02309 | 0.02233 | 0.00971 | 0.01320 | | | | | | | | | | |
| 7 | TGEV AHHF | 0.01253 | 0.01886 | 0.01953 | 0.01461 | 0.01953 | 0.01251 | | | | | | | | | |
| 8 | TGEV H16 | 0.01955 | 0.02523 | 0.02447 | 0.01321 | 0.01742 | 0.00902 | 0.01111 | | | | | | | | |
| 9 | PRCV HOL87 | 0.03072 | 0.03413 | 0.03658 | 0.02983 | 0.02899 | 0.02563 | 0.02647 | 0.02817 | | | | | | | |
| 10 | PRCV 135 (86/135308) | 0.03241 | 0.03582 | 0.03827 | 0.02983 | 0.02983 | 0.02983 | 0.02901 | 0.02901 | 0.01067 | | | | | | |
| 11 | PRCV 137 (86/137004) | 0.03156 | 0.03497 | 0.03743 | 0.02899 | 0.02815 | 0.02479 | 0.02733 | 0.02733 | 0.00902 | 0.00655 | | | | | |
| 12 | PRCV AR310 | 0.02406 | 0.02913 | 0.02906 | 0.01817 | 0.02151 | 0.01650 | 0.01985 | 0.01902 | 0.01902 | 0.02990 | 0.02990 | | | | |
| 13 | PRCV ISU-1 | 0.02322 | 0.02744 | 0.02822 | 0.01733 | 0.02067 | 0.01567 | 0.01567 | 0.01818 | 0.02993 | 0.03077 | 0.02993 | 0.00657 | | | |
| 14 | PRCV OH7269 | 0.03643 | 0.04153 | 0.04228 | 0.03049 | 0.03301 | 0.02882 | 0.03219 | 0.03136 | 0.03853 | 0.03939 | 0.03853 | 0.01824 | 0.01990 | | |
| 15 | TGEV/USA/Minnesota138/2006 | 0.02946 | 0.03591 | 0.03512 | 0.02374 | 0.02799 | 0.02304 | 0.02659 | 0.02588 | 0.03404 | 0.03489 | 0.03404 | 0.00987 | 0.01070 | 0.02050 | |
| 16 | TGEV 96-1933 | 0.05907 | 0.06279 | 0.06339 | 0.05169 | 0.05679 | 0.04952 | 0.05537 | 0.05391 | 0.05194 | 0.05538 | 0.05366 | 0.04263 | 0.04178 | 0.05678 | 0.05533 |

**Note:** a distance matrix was built using the Poisson correction model based on the results of multiple alignment of S-protein amino acid sequences performed according to the MUSCLE algorithm.

135 and PRCV 137, PRCV ISU-1 and PRCV AR310, PRCV HOL87 and PRCV 137, TGEV Miller M6 and TGEV H16, as well as PRCV AR310 and TGEV/USA/Minnesota138/2006. Conversely, the strain TGEV 96–1933 demonstrates the most substantial distances with all other strains.

## Assessment of the stability of RBD-APN complexes and the impact of mutations

To determine the strains of TGEV and PRCV with the highest affinity for APN receptors, ΔG calculations were conducted. This parameter signifies the energy released during the virus-receptor binding process, thus indicating the stability of the resulting RBD-APN complexes. More negative ΔG values indicate greater stability of these complexes. The results of ΔG calculations for models of complexes, which indicate bound RBDs of various TGEV and PRCV strains with pig and human receptors, are presented in Tables 2 and 3. In all cases described below, the data followed a normal distribution; however, due to the presence of several outliers, nonparametric methods were employed to evaluate the differences between groups.

The range of ΔG values obtained for pAPN complexes with the RBD of various TGEV strains spans −53.72–−63.51 kcal/mol, while for pAPN complexes with PRCV strains, the range extends −54.85–−60.37 kcal/mol. An outlier was identified within the TGEV group (TGEV USA/Minnesota138/2006 strain) at the upper bound of the ΔG range (−53.72 kcal/mol). The differences between the ΔG values for complexes containing the TGEV and PRCV strain groups are not statistically significant ($p = 0.957$), suggesting a comparable affinity for the pig receptor among RBDs of both TGEV and PRCV strains.

Concerning the putative complexes formed between viral RBDs and hAPN, the range of ΔG values extends −52.80–−60.68 kcal/mol for TGEV strains and −52.79–−57.58 kcal/mol for PRCV strains. Outliers were identified within the TGEV group for the following strains: TGEV USA/Minnesota138/2006 (−52.80 kcal/mol), TGEV TO14 (−58.57 kcal/mol), and TGEV 133 (−60.68 kcal/mol). The difference in ΔG values between complexes formed by groups of TGEV and PRCV strains with the human receptor cannot be considered statistically significant ($p = 0.664$).

Comparing the ΔG values for complexes formed by the same strains of viruses with pAPN and hAPN, it is possible to conclude that ΔG tends to be more negative for RBD-pAPN complexes than for RBD-hAPN complexes. This observation suggests that viruses tend to form more stable complexes with the pig receptor than with the human receptor, which aligns with expectations since the pig is the host for TGEV and PRCV. The differences in ΔG of complexes formed by TGEV

**Table 2. ΔG (kcal/mol) calculations for complexes of receptor-binding domains of TGEV strains with APN receptors.**

|  | TGEV Purdue (1447 aa) | TGEV SZ19 (1446 aa) | TGEV 96–1933 (1449 aa) | TGEV 133 (1449 aa) | TGEV TO14 (1449 aa) | TGEV FS772/70 (1449 aa) | TGEV Miller M6 virulent (1449 aa) | TGEV AHHF (1448 aa) | TGEV H16 (1448 aa) | TGEV USA/Minnesota138/2006 (1449 aa) |
|---|---|---|---|---|---|---|---|---|---|---|
| Pig | −57.83 | −57.59 | −61.90 | −63.51 | −60.23 | −58.19 | −57.78 | −57.68 | −57.78 | −53.72 |
| Human | −55.76 | −56.01 | −54.89 | −60.68 | −58.57 | −55.40 | −55.09 | −55.35 | −55.29 | −52.80 |
| ΔΔG | 2.07 | 1.58 | 7.01 | 2.83 | 1.66 | 2.79 | 2.69 | 2.33 | 2.49 | 0.92 |

**Note.** In the context of this Table, ΔΔG represents the change in ΔG values for human complexes in comparison to the corresponding pig complexes.

**Table 3. ΔG (kcal/mol) calculations for complexes of receptor-binding domains of PRCV strains with APN receptors, kcal/mol.**

|  | PRCV-HOL-87 (1225 aa) | PRCV 135 (1225 aa) | PRCV 137 (1225 aa) | PRCV AR310 (1222 aa) | PRCV ISU-1 (1222 aa) | PRCV OH7269 (1232 aa) |
|---|---|---|---|---|---|---|
| Pig | −58.18 | −60.37 | −58.04 | −54.85 | −55.88 | −60.15 |
| Human | −55.25 | −57.58 | −55.40 | −52.79 | −53.37 | −56.89 |
| ΔΔG | 2.93 | 2.79 | 2.64 | 2.06 | 2.51 | 3.26 |

**Note.** Q.v. Table 2.

strains with pig and human receptors are statistically significant ($p = 0.00195$), as are the differences in ΔG of complexes formed by PRCV strains with pig and human receptors ($p = 0.0313$). Thus, both TGEV and PRCV virus strains examined demonstrate reduced affinity for hAPN compared to pAPN.

The results presented in Table 2 and Table 3 indicate that the RBD of the TGEV 133 strain exhibits the greatest stability with both pAPN and hAPN among all the strains studied. This strain possesses its own specific mutational profile, distinguishing it from Purdue by three mutations: H562N (H560N), N563K (N561K), and D600H (D598H) (hereinafter, in parentheses, the position of the mutation is in relation to the Purdue strain, which has a double amino acid deletion in the S-protein). This mutational profile renders the TGEV 133 strain the most affine for both pig and human receptors.

It was assessed whether an additional mutation could further enhance the affinity of the virus for receptors beyond that recorded for the TGEV 133 strain. For this purpose, over 500 mutations (S3 Table, S4 Table) were analyzed, which can arise in the RBD region as a result of single-nucleotide substitutions in codons, leading to corresponding amino acid substitutions in the RBD. Single-nucleotide mutations in this article are regarded as the most probable missense variants from an evolutionary perspective. The RBD sequence from the TGEV 133 strain served as the basis for this analysis, as it already contains several important mutations that stabilize the RBD-APN complex.

Table 4 and Table 5 summarize estimates of the change in free energy (ΔΔG, kcal/mol) of binding to the pAPN and hAPN receptors, respectively, as a consequence of mutations in the RBD. Estimates are provided only for those mutations for which a stabilizing (or at least neutral) effect has been confirmed by the majority of services for humans or pigs. Estimates for all mutations studied are presented in S3 Table and S4 Table.

To more accurately compare the stability of complexes involving APN receptors and mutated RBDs with the primary RBD-APN complexes for the TGEV 133 strain, the MM/GBSA approach was once again applied, and ΔG values were calculated. The results obtained are also presented in Tables 4 and 5. It was found that the most stable complexes are observed with both pig and human receptors, provided that mutations P547T (P545T), T551I (T549I), or T551K (T549K) are present in the RBD of the virus.

To assess the impact of multiple mutations on the affinity of the RBD to both pig and human APN receptors, simultaneous introduction of pairs of stabilizing mutations P547T+T551I and P547T+T551K was performed, also using the RBD sequence of the TGEV 133 strain as a basis. The ΔG was then calculated employing the MM/GBSA approach (Table 6). Results revealed that the combination of mutations P547T+T551K significantly enhances the stability of complexes with both pAPN and hAPN receptors. In this case, the ΔG values are lower than those observed for any other complex containing the RBD of a known strain or an RBD with single mutations.

Fig 3A presents a visualization of the overall three-dimensional structure of the RBD-APN complex, highlighting the positions where the TGEV 133 strain has mutations relative to the reference Purdue strain, as well as the locations of theoretical mutations predicted to enhance the virus's affinity for the APN receptor. Fig 3B–C also illustrates the interfaces of the RBD of TGEV 133 with pAPN and hAPN, highlighting the amino acid residues involved in hydrogen bond formation. On the RBD side, these contacts with both pAPN and hAPN are formed by G543, Y544, Q546, and W587. Additionally, A549 contributes to a contact with pAPN, while R541 forms a contact with hAPN. Notably, the theoretical substitutions analyzed at positions 547 and 551 are in close proximity to these contact-forming residues. S5 Fig schematically illustrates the contacts, including H-bonds and non-bonded interactions, at the interfaces of pAPN and hAPN complexes with the RBDs of different TGEV strains.

## Molecular dynamics of RBD-APN complexes

Molecular dynamics simulations were conducted for complexes of both pAPN and hAPN with the RBDs of three TGEV strains: Purdue (serving as the reference strain), 133 (identified as having the highest receptor affinity), and a theoretical strain derived from 133 with two additional stabilizing mutations, P547T+T551K. A total of six 400-ns simulations were performed. The RMSD plots corresponding to the molecular dynamics trajectories of these six RBD-APN complexes are

**Table 4. Prediction on the ΔΔG and ΔG (kcal/mol) of the complexes, involving pAPN and receptor-binding domain of the TGEV 133 with single mutations.**

| Amino acid substitution | Original codone | ΔΔG | | | | | ΔG (MM/GBSA) |
|---|---|---|---|---|---|---|---|
| | | mCSM-PPI2 | BindProfX | SAAMBE-3D | BeAtMuSIC | MutaBind2 | |
| P523T | CCT | −0.033 | — | −0.040 | −0.060 | 0.590 | −63.63 |
| P547L | CCC | 0.266 | 0.000 | −0.100 | −0.280 | 0.490 | −63.36 |
| P547S | | 0.096 | 0.000 | 0.230 | 0.160 | 0.530 | −64.09 |
| P547T | | −0.227 | 0.000 | −0.130 | −0.240 | 0.520 | **−64.36** |
| A549D | GCC | −0.263 | 1.207 | −1.230 | 0.780 | 2.000 | −60.32 |
| A549S | | −0.100 | 0.956 | −0.330 | 0.250 | 0.520 | −63.13 |
| A549T | | −0.139 | 1.076 | −0.390 | 0.390 | 0.280 | −63.27 |
| T551I | ACA | −0.003 | 0.964 | −0.050 | 0.510 | −0.020 | **−66.35** |
| T551K | | −0.109 | 0.985 | −0.080 | 0.620 | 0.040 | **−65.13** |
| T551P | | −0.158 | 1.130 | 0.360 | −0.170 | −0.280 | −62.29 |
| L552F | TTA | −0.205 | 0.000 | −0.260 | 0.050 | 0.210 | −63.78 |
| M559I | ATG | −0.031 | — | 0.280 | 0.000 | −0.080 | −63.52 |
| M559L | | −0.016 | — | 0.290 | −0.060 | −0.100 | −63.65 |
| M559V | | −0.093 | — | 0.390 | −0.080 | −0.140 | −63.6 |
| S580Y | TCT | −0.048 | — | 0.220 | −0.320 | −0.400 | −63.66 |
| A585D | GCT | −0.451 | 0.000 | −0.850 | 0.410 | 0.950 | −58.59 |
| A585S | | −0.298 | 0.000 | −0.050 | 0.260 | −0.060 | −63.33 |
| A585T | | −0.222 | 0.000 | −0.100 | 0.200 | 0.290 | −63.3 |
| I590F | ATT | −0.254 | 0.824 | 0.040 | 0.280 | 0.100 | −63.36 |
| K592R | AAG | −0.077 | — | 0.350 | 0.000 | −0.460 | −64.08 |
| R593Q | CGA | −0.154 | 0.000 | 0.770 | 0.070 | 0.180 | −59.73 |
| N594D | AAC | −0.163 | — | −0.010 | −0.010 | 0.220 | −60.17 |
| N594H | | −0.055 | — | −0.030 | 0.000 | 0.160 | −63.43 |
| L599F | TTA | −0.629 | 0.000 | −0.190 | 0.380 | 0.040 | −63.67 |
| A603V | GCT | −0.052 | — | −0.250 | −0.020 | −0.100 | −63.59 |
| K616R | AAA | −0.033 | — | 0.230 | −0.070 | −0.080 | −63.63 |
| K616N | | −0.072 | — | 0.150 | −0.150 | −0.020 | −61.46 |
| L617F | TTG | −0.321 | — | −0.090 | −0.010 | 0.100 | −63.48 |
| V633F | GTT | −0.208 | — | −0.170 | 0.100 | −0.050 | −63.55 |
| A635T | GCT | −0.136 | — | −0.090 | −0.050 | 0.210 | −63.63 |
| A642S | GCT | −0.099 | — | −0.410 | 0.250 | −0.030 | −63.56 |
| A642T | | −0.266 | — | −0.460 | 0.140 | 0.150 | −63.48 |
| A642V | | −0.080 | — | 0.020 | −0.030 | −0.520 | −63.65 |
| V651F | GTT | −0.291 | 0.000 | −0.190 | 0.140 | 0.130 | −63.59 |
| N664D | AAC | −0.249 | — | −0.060 | −0.030 | 0.270 | −61.67 |
| P669Q | CCG | −0.220 | — | −0.040 | −0.200 | 1.120 | −63.54 |

**Note.** ΔΔG values obtained using mCSM-PPI2 are presented with the opposite sign to align them with values from other tools. Table cells with ΔΔG values indicating a stabilizing effect from mutations are shaded in light blue, while those indicating a destabilizing effect are shaded in purple. A dash in a cell indicates that the tool does not calculate the effect of the mutation.

**Table 5. Prediction on the ΔΔG and ΔG (kcal/mol) of the complexes, involving hAPN and receptor-binding domain of the TGEV 133 with single mutations.**

| Amino acid substitution | Original codone | ΔΔG | | | | | ΔG (MM/GBSA) |
|---|---|---|---|---|---|---|---|
| | | mCSM-PPI2 | BindProfX | SAAMBE-3D | BeAtMuSIC | MutaBind2 | |
| P523T | CCT | −0.046 | — | −0.040 | −0.060 | 0.580 | −60.64 |
| P547L | CCC | 0.329 | 0.000 | −0.300 | −0.390 | 0.910 | −62.93 |
| P547S | | 0.016 | 0.000 | −0.050 | −0.020 | 0.500 | −60.35 |
| P547T | | −0.276 | 0.000 | −0.260 | −0.300 | 0.750 | **−63.4** |
| A549D | GCC | −0.278 | 0.000 | −0.970 | 0.550 | 0.990 | −58.43 |
| A549S | | −0.022 | 0.000 | −0.170 | 0.300 | 0.190 | −60.5 |
| A549T | | −0.091 | 0.000 | −0.220 | 0.230 | 0.140 | −60.99 |
| T551I | ACA | −0.031 | 0.964 | −0.050 | 0.650 | 0.280 | **−63.41** |
| T551K | | −0.039 | 0.985 | −0.080 | 0.450 | −0.150 | **−65.83** |
| T551P | | −0.142 | 1.130 | 0.360 | −0.140 | −0.600 | −60.99 |
| L552F | TTA | −0.182 | 1.764 | −0.260 | 0.030 | 0.020 | −60.48 |
| M559I | ATG | −0.014 | — | 0.280 | 0.000 | −0.100 | −60.67 |
| M559L | | 0.016 | — | 0.290 | −0.060 | −0.130 | −60.68 |
| M559V | | −0.052 | — | 0.390 | −0.080 | −0.070 | −60.89 |
| S580Y | TCT | −0.049 | — | 0.220 | −0.310 | −0.440 | −60.25 |
| A585D | GCT | −0.454 | 0.000 | −0.850 | 0.400 | 1.050 | −57.97 |
| A585S | | −0.289 | 0.000 | −0.050 | 0.260 | −0.060 | −61.0 |
| A585T | | −0.284 | 0.000 | −0.100 | 0.170 | 0.400 | −60.56 |
| I590F | ATT | −0.210 | 0.824 | −0.080 | 0.280 | −0.080 | −60.29 |
| K592R | AAG | −0.069 | — | 0.350 | 0.010 | −0.330 | −60.87 |
| R593Q | CGA | −0.159 | 0.000 | 0.770 | −0.010 | 0.170 | −58.11 |
| N594D | AAC | −0.182 | — | −0.010 | −0.010 | 0.160 | −58.19 |
| N594H | | −0.059 | — | −0.030 | −0.010 | 0.190 | −60.84 |
| L599F | TTA | −0.616 | 0.000 | −0.190 | 0.290 | 0.780 | −61.03 |
| A603V | GCT | −0.041 | — | −0.250 | −0.020 | −0.200 | −60.82 |
| K616R | AAA | −0.020 | — | 0.230 | −0.070 | −0.010 | −60.88 |
| K616N | | −0.039 | — | 0.150 | −0.150 | 0.050 | −59.03 |
| L617F | TTG | −0.314 | — | −0.090 | −0.010 | 0.080 | −60.65 |
| V633F | GTT | −0.195 | — | −0.170 | 0.120 | −0.060 | −60.72 |
| A635T | GCT | −0.126 | — | −0.090 | −0.080 | 0.180 | −60.71 |
| A642S | GCT | −0.093 | — | −0.410 | 0.230 | 0.220 | −60.72 |
| A642T | | −0.263 | — | −0.046 | 0.140 | 0.200 | −60.7 |
| A642V | | −0.096 | — | 0.020 | −0.010 | −0.540 | −60.8 |
| V651F | GTT | −0.314 | — | −0.190 | 0.120 | 0.120 | −60.87 |
| N664D | AAC | −0.216 | — | −0.060 | −0.010 | 0.200 | −58.98 |
| P669Q | CCG | −0.135 | — | −0.040 | −0.180 | 0.970 | −60.91 |

**Note.** Q.v. Table 4.

**Table 6. ΔG (kcal/mol) calculations for complexes involving TGEV 133 receptor-binding domain with double mutations.**

| Double mutations | pAPN | hAPN |
| --- | --- | --- |
| P547T+T551I | −65.69 | −62.38 |
| P547T+T551K | **−70.96** | **−67.53** |

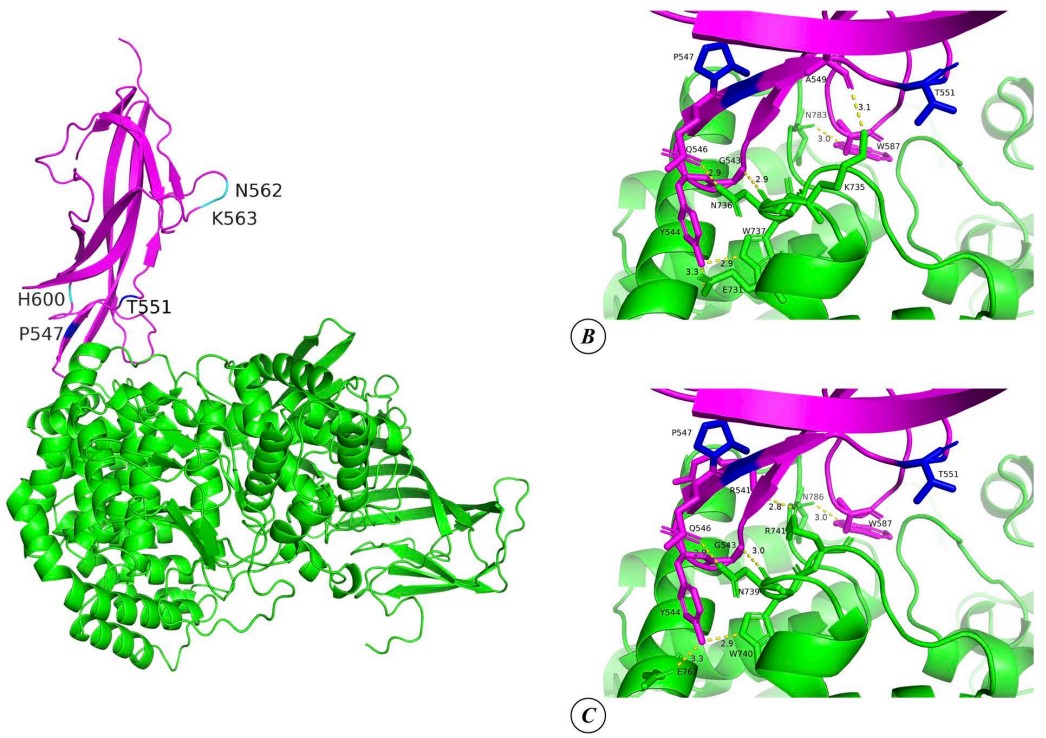

**Fig 3. Three-dimensional structure of the RBD of TGEV 133 strains in complexes with pAPN and hAPN: general view of complex with pAPN (A), interfaces of the complexes with pAPN (B) and hAPN (C).** The structure of APN is depicted in green, while the RBD is shown in magenta. The positions of mutations in the TGEV 133 strain relative to the reference Purdue strain are highlighted in cyan, and the positions of theoretical mutations that may stabilize the complex are indicated in blue. Length of H-bonds is indicated in angstrom (Å).

shown in Fig 4A–B. Over the course of the simulations, the dynamics of all complexes demonstrated the reduction in RMSD fluctuations. For this reason, RMSF indices for C-alpha atoms were calculated based on the final 100 ns of the molecular dynamics trajectories.

The RMSF plots for pAPN and hAPN are presented in Fig 4C–D. Both the porcine and human receptors display a similar trend in terms of C-alpha atom fluctuations upon interaction with the RBDs of different TGEV strains. For pAPN, the RMSF values are very similar for complexes with the Purdue and 133 strains, while they are comparatively higher for the complex involving the theoretically mutated 133 strain. In the case of hAPN, the RMSF values are also close for the Purdue and 133 complexes; however, the RMSF values are comparatively lower for the receptor in the complex with the mutated 133 strain.

The RMSF plot for RBDs of the different TGEV strains complexed with pAPN (Fig 4E) reveals that the RBD of the mutated 133 strain exhibits the largest fluctuations, whereas the RBDs of the Purdue and 133 strains display similar RMSF values. In contrast, the RMSF analysis of RBDs complexed with hAPN (Fig 4F) shows that the Purdue strain experiences the largest fluctuations.

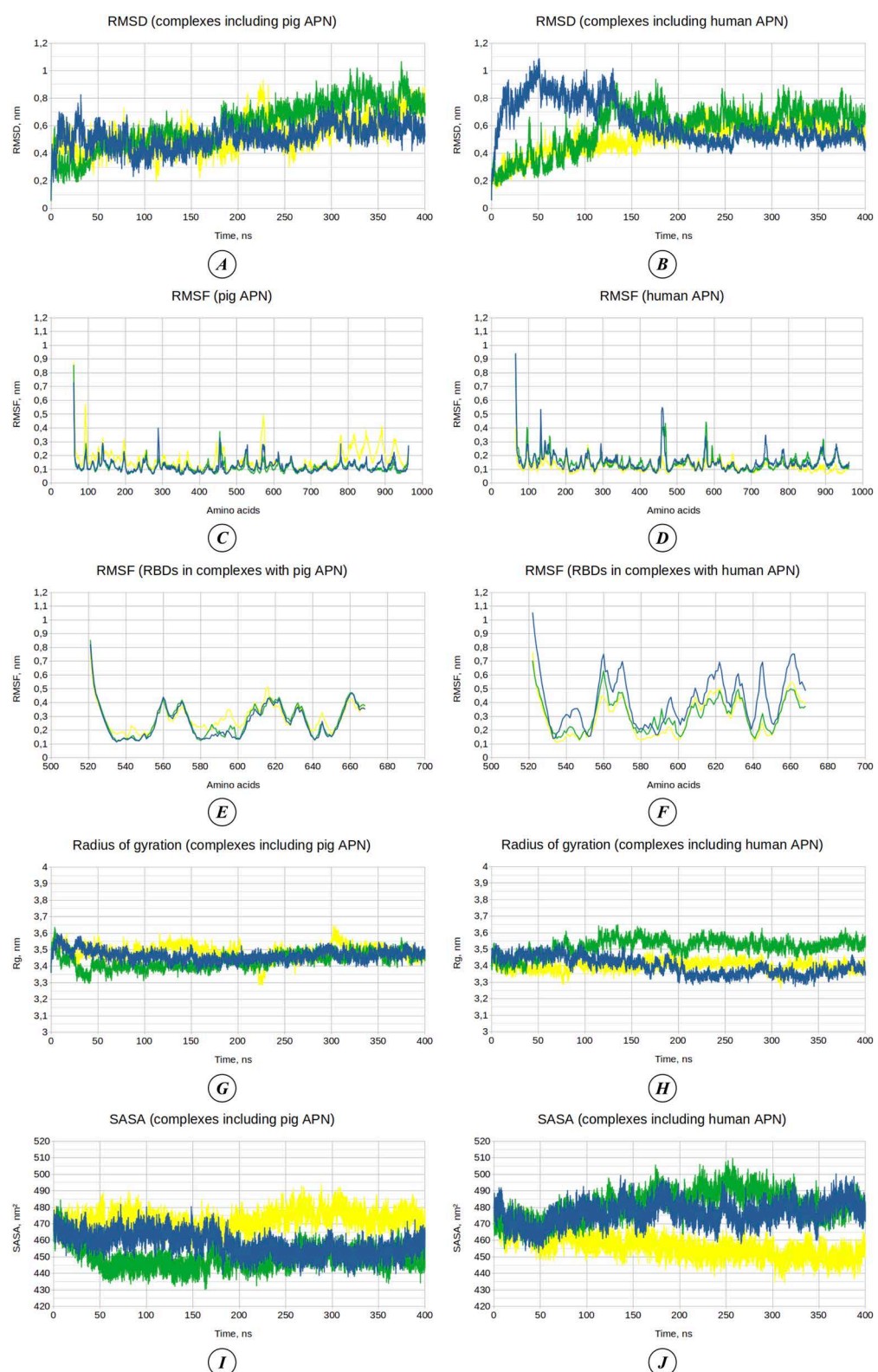

**Fig 4. Analysis of molecular dynamics trajectories. (A–B)** Root mean square deviation (RMSD) of complexes involving pAPN **(A)** and hAPN **(B)**. **(C–D)** Root mean square fluctuation (RMSF) of pAPN **(C)** and hAPN **(D)**. **(E–F)** RMSF of RBDs of virus strains in complexes involving pAPN **(E)** and

hAPN **(F)**. **(G–H)** Radius of gyration (Rg) of complexes involving pAPN **(G)** and hAPN **(H)**. **(I–J)** Solvent-accessible surface area (SASA) of complexes involving pAPN **(I)** and hAPN **(J)**. Graphs for complexes involving the TGEV Purdue RBD are shown in blue, the TGEV 133 RBD in green, and the TGEV 133 RBD with additional mutations in yellow. Residue numbering in panels **(E)** and **(F)** follows that of the Purdue strain. The raw data used in the analysis of RMSD, RMSF, Rg, and SASA are provided in S6 Table.

The radius of gyration values, which reflect the compactness of the molecules, were relatively similar for all studied complexes and remained fairly uniform throughout the simulations (Fig 4G–H). For the complexes involving pAPN, the average radius of gyration over the entire simulation time was $3.47 \pm 0.03$ nm, $3.43 \pm 0.04$ nm, and $3.49 \pm 0.04$ nm when interacting with the RBDs of the Purdue strain, 133 strain, and 133 strain with P547T+T551K mutations, respectively. For the complexes involving hAPN, the radius of gyration was $3.40 \pm 0.05$ nm, $3.52 \pm 0.05$ nm, and $3.40 \pm 0.03$ nm for interactions with the RBDs of the Purdue strain, 133 strain, and mutated 133 strain, respectively.

The solvent-accessible surface area (SASA), which reflects the diffuseness of the molecular structures, is shown in Fig 4I–J. For the complexes involving pAPN, the average SASA values over the entire simulation time were $457.58 \pm 6.66$ nm², $449.57 \pm 6.51$ nm², and $472.18 \pm 5.49$ nm² for interactions with the RBDs of the Purdue strain, 133 strain, and mutated 133 strain, respectively. For the complexes involving hAPN, the average SASA values over the entire simulation time were $476.62 \pm 6.70$ nm², $481.19 \pm 8.38$ nm², and $456.26 \pm 6.81$ nm² for interactions with the RBDs of the Purdue strain, 133 strain, and mutated 133 strain, respectively. Differences were observed between the interactions with human and porcine receptors: the highest SASA values were associated with the complex involving pAPN and the RBD of the mutated 133 strain, while for the complexes involving hAPN, the interaction with the RBD of the mutated 133 strain resulted in the lowest SASA values.

The dynamics of the number of H-bonds formed between the receptors and the RBDs of the viruses are shown in Fig 5A–B. The average number of H-bonds for pAPN complexed with the Purdue, 133, and mutated 133 strains was $5.81 \pm 1.76$, $7.36 \pm 2.20$, and $7.93 \pm 2.37$, respectively. For hAPN complexed with the Purdue, 133, and mutated 133 strains, the average number of H-bonds was $4.09 \pm 2.27$, $3.80 \pm 2.09$, and $3.70 \pm 2.18$, respectively. Thus, the number of H-bonds formed during interactions with pAPN was consistently greater than those formed with hAPN, regardless of the strain studied. Additionally, the RBDs of the 133 strain and its variant with two additional theoretical mutations formed a greater number of H-bonds with pAPN compared to the reference Purdue strain.

The number of non-polar contacts (Fig 5C–D) is significantly greater than that of H-bonds, which can be attributed to the less stringent criteria for their definition (which include all non-hydrogen atom contacts between APN and RBD within a distance of 0.6 nm). The average number of non-polar contacts was $846.68 \pm 74.73$, $857.43 \pm 110.31$, and $1001.02 \pm 123.26$ for pAPN complexed with the RBDs of the Purdue, 133, and mutated 133 strains, respectively. For hAPN complexed with the same RBDs, the average number of non-polar contacts was $903.23 \pm 187.32$, $732.24 \pm 122.54$, and $923.72 \pm 101.17$, respectively. In this case, the number of non-polar contacts is greater for complexes with pAPN compared to hAPN only when interacting with the 133 and mutated 133 strains. The RBD of the Purdue strain, on the contrary, forms more non-polar contacts when interacting with hAPN. When comparing different TGEV strains, it may be concluded that the greatest number of non-polar contacts is formed by the 133 strain with two additional mutations; this pattern is consistent for both human and porcine receptors. This highlights the impact of the introduced mutations (P547T and T551K) in enhancing receptor binding interactions.

After molecular dynamics simulations, binding free energy was calculated to estimate changes in the affinity of the viruses for the porcine and human receptors over 400 ns. The total binding free energy ($\Delta G_{total}$) (Fig 5E–F) is presented as the sum of two main components: molecular mechanics energy in the gas phase ($\Delta G_{gas}$) and solvation energy ($\Delta G_{solv}$), which are shown in Fig 5G–J. At the same time, $\Delta G_{gas}$ and $\Delta G_{solv}$ each consist of several other energy components, which is reflected in Table 7. The dynamics of changes in van der Waals interaction energy ($\Delta E_{vdW}$) and electrostatic interaction energy ($\Delta E_{eel}$), which contribute to changes in $\Delta G_{gas}$, are shown in Fig 6A-D. The dynamics of changes in polar solvation energy ($\Delta E_{pb}$) and non-polar solvation energy ($\Delta E_{npolar}$), which contribute to $\Delta G_{solv}$, are presented in Fig 6E-H.

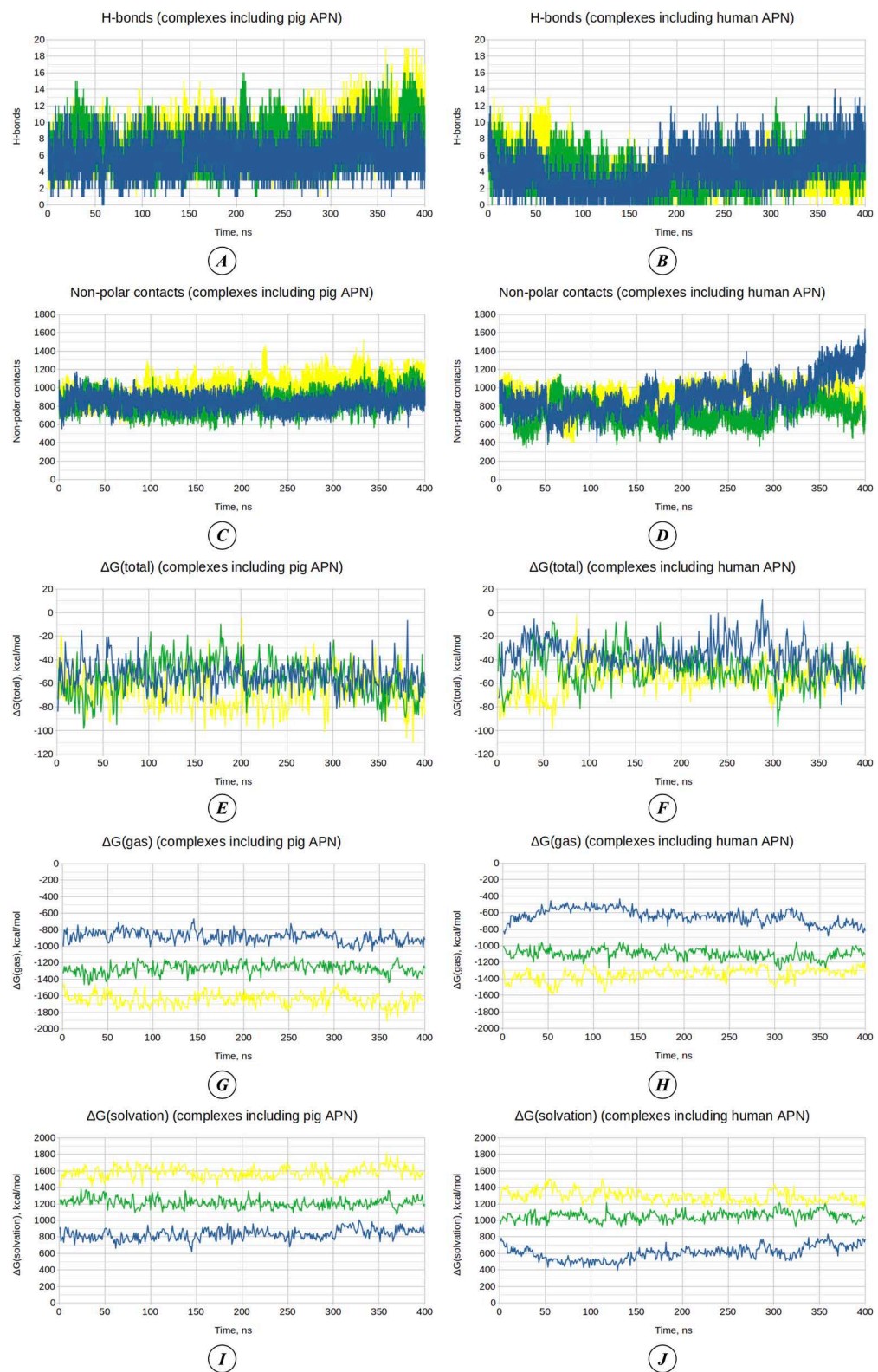

**Fig 5. Analysis of virus binding to APN through molecular dynamics. (A–B)** H-bonds between RBD and APN in complexes involving pAPN **(A)** and hAPN **(B)**. **(C–D)** Non-polar contacts between RBD and APN in complexes involving pAPN **(C)** and hAPN **(D)**. **(E–F)** Molecular mechanics energy in the

gas phase (ΔG$_{gas}$) for complexes involving pAPN **(E)** and hAPN **(F)**. **(G–H)** Solvation energy (ΔG$_{solv}$) for complexes involving pAPN **(G)** and hAPN **(H)**. **(I–J)** Binding free energy (ΔG$_{total}$) for complexes involving pAPN **(I)** and hAPN **(J)**. Graphs for complexes involving the TGEV Purdue RBD are shown in blue, the TGEV 133 RBD in green, and the TGEV 133 RBD with additional mutations in yellow. The raw data used in the analysis of H-bonds, non-polar contacts, and the components of binding free energy are provided in S6 Table and S7 Table.

The average values of ΔG$_{total}$ calculated for the complexes of pAPN with the Purdue, 133, and mutated 133 strains are −53.70 ± 11.44 kcal/mol, −55.85 ± 15.52 kcal/mol, and −66.75 ± 15.36 kcal/mol, respectively. These results confirm that the 133 strain of TGEV has a higher affinity for pAPN than the reference Purdue strain, while the 133 strain with additional theoretical P547T+T551K mutations forms the most stable complex. Similar patterns are observed for interactions with hAPN: the average ΔG$_{total}$ values are −36.02 ± 13.49 kcal/mol, −48.34 ± 14.00 kcal/mol, and −55.48 ± 13.29 kcal/mol for complexes with the Purdue, 133, and mutated 133 strains, respectively.

It is important to note that the complexes of the Purdue and 133 strains with the human receptor exhibit higher ΔG$_{total}$ values than their corresponding complexes with the porcine receptor, indicating reduced affinity for the human receptor. However, the ΔG$_{total}$ values for hAPN complexed with the mutated 133 strain are comparable to the values for pAPN interactions with TGEV RBDs. These findings indicate that point mutations at specific positions in the RBD of TGEV can both enhance the virus's ability to interact with the porcine receptor and increase its affinity for the human receptor.

PCA and FEL analyses based on the C-alpha atoms of RBD-APN complexes reveal distinct differences between complexes containing different sets of mutations in the RBD (Fig 7). The PCA projection of the positional fluctuations of C-alpha atoms for complexes involving pAPN onto the first two principal components, which account for 48.84% and 21.09% of the structural variability observed during molecular dynamics, is shown in Fig 7A. The coordinates of the plot points corresponding to pAPN complexes with the RBD of TGEV Purdue, TGEV 133, and TGEV 133 with additional mutations outline regions on the PCA plot that partially overlap. Individual PCA plots for each complex, showing data projected onto the same principal component system but displayed without overlap, are provided in Fig 7B–D. These separate plots

**Table 7. Binding free energy (ΔG, kcal/mol) and its components calculated from conformational states of RBD-APN complexes along molecular dynamics trajectories.**

| ΔG elements | TGEV Purdue/ pAPN | TGEV 133/ pAPN | TGEV 133 mutated/ pAPN | TGEV Purdue/ hAPN | TGEV 133/ hAPN | TGEV 133 mutated/ hAPN |
|---|---|---|---|---|---|---|
| ΔE$_{bonded}$ | 0.00 | 0.00 | 0.00 | 0.00 | 0.00 | 0.00 |
| ΔE$_{vdW}$ | −61.19 ± 5.49 | −60.64 ± 7.42 | −70.62 ± 8.66 | −72.47 ± 13.24 | −57.99 ± 8.50 | −72.23 ± 8.36 |
| ΔE$_{eel}$ | −824.35 ± 66.09 | −1207.21 ± 61.78 | −1576.30 ± 75.28 | −568.08 ± 78.16 | −1041.57 ± 57.57 | −1274.28 ± 77.38 |
| ΔE$_{nonbonded}$ (ΔE$_{vdW}$ + ΔE$_{eel}$) | −885.54 ± 66.71 | −1267.85 ± 62.81 | −1646.92 ± 75.70 | −640.55 ± 86.53 | −1099.56 ± 59.89 | −1346.51 ± 75.43 |
| ΔE$_{pb}$ | 840.16 ± 62.80 | 1220.75 ± 56.22 | 1590.37 ± 71.86 | 614.22 ± 84.92 | 1059.42 ± 55.81 | 1299.92 ± 70.92 |
| ΔE$_{npolar}$ | −8.32 ± 0.46 | −8.74 ± 0.67 | −10.19 ± 0.95 | −9.69 ± 1.73 | −8.20 ± 0.83 | −8.90 ± 0.56 |
| ΔE$_{disper}$ | 0.00 | 0.00 | 0.00 | 0.00 | 0.00 | 0.00 |
| ΔG$_{gas}$ (ΔE$_{bonded}$ + ΔE$_{nonbonded}$) | −885.54 ± 66.71 | −1267.85 ± 62.81 | −1646.92 ± 75.70 | −640.55 ± 86.53 | −1099.56 ± 59.89 | −1346.51 ± 75.43 |
| ΔG$_{solv}$ (ΔE$_{pb}$ + ΔE$_{npolar}$ + ΔE$_{disper}$) | 831.84 ± 62.59 | −1212.00 ± 56.00 | 1580.17 ± 71.52 | 604.53 ± 83.66 | 1051.22 ± 55.38 | 1291.02 ± 70.69 |
| ΔG (ΔG$_{gas}$ + ΔG$_{solv}$) | −53.70 ± 11.44 | −55.85 ± 15.52 | −66.75 ± 15.36 | −36.02 ± 13.49 | −48.34 ± 14.00 | −55.48 ± 13.29 |

**Note:** ΔE$_{bonded}$ – bonded interaction energy, ΔE$_{vdW}$ – van der Waals interaction energy, ΔE$_{eel}$ – electrostatic interaction energy, ΔE$_{nonbonded}$ – non-bonded interaction energy, ΔE$_{pb}$ – polar solvation energy calculated using Poisson-Boltzmann method, ΔE$_{npolar}$ – non-polar solvation energy, ΔE$_{disper}$ – dispersion interaction energy, ΔG$_{gas}$ – gas-phase binding free energy, ΔG$_{solv}$ – solvation free energy, ΔG – total binding free energy. The raw data used in the analysis of the components of binding free energy are provided in S7 Table.

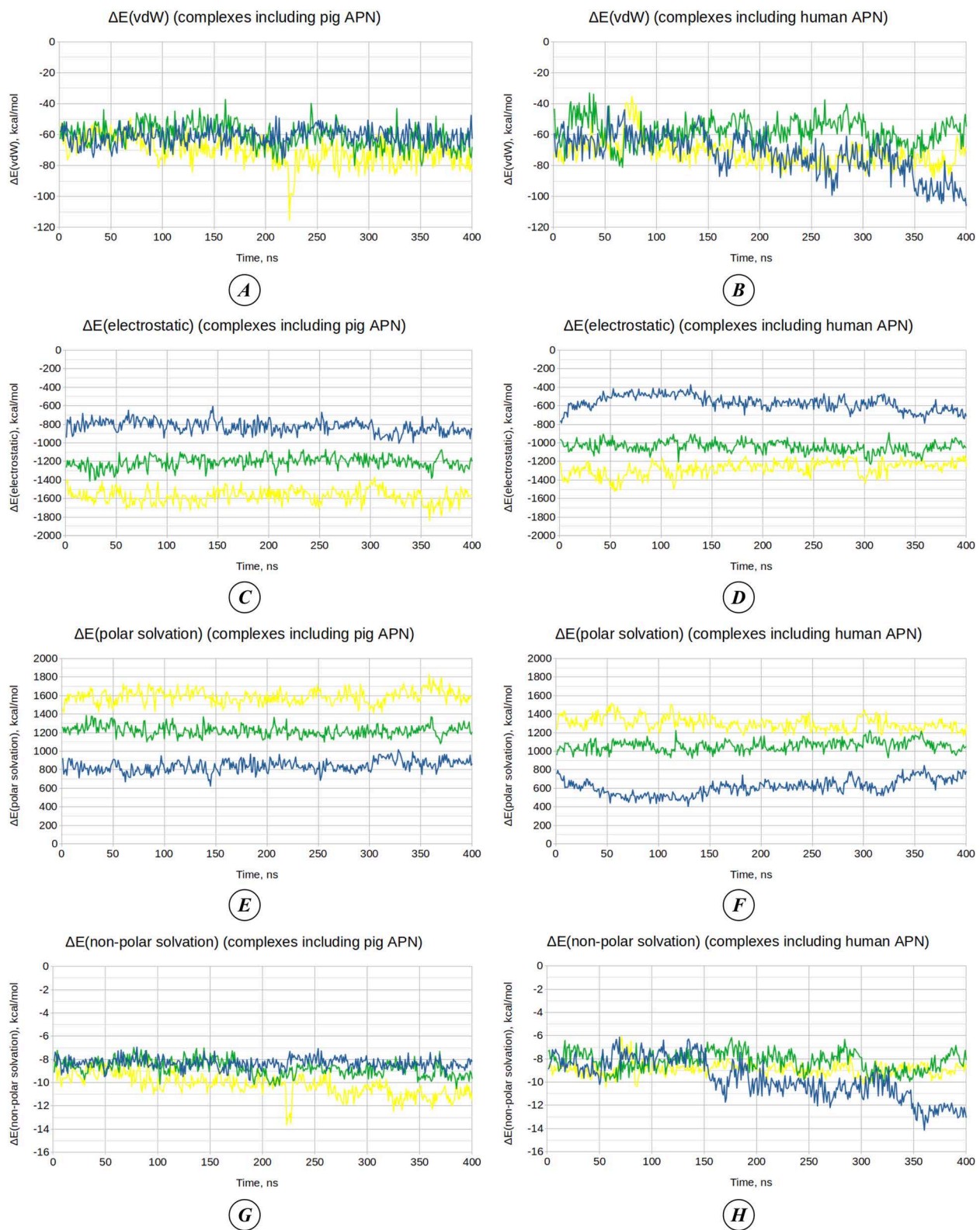

**Fig 6. Dynamics of energy components (ΔE, kcal/mol) that contribute to the binding free energy. (A–B)** Van der Waals interaction energy ($\Delta E_{vdW}$) for complexes involving pAPN **(A)** and hAPN **(B)**.**(C–D)** Electrostatic interaction energy ($\Delta E_{eel}$) for complexes involving pAPN **(C)** and hAPN **(D)**. **(E–F)** Polar solvation energy calculated using Poisson-Boltzmann method ($\Delta E_{pb}$) for complexes involving pAPN **(E)** and hAPN **(F)**. **(G–H)** Non-polar solvation

energy (ΔE$_{npolar}$) for complexes involving pAPN **(G)** and hAPN **(H)**. Graphs for complexes involving the TGEV Purdue RBD are shown in blue, the TGEV 133 RBD in green, and the TGEV 133 RBD with additional mutations in yellow. The raw data used in the analysis of the components of binding free energy are provided in S7 Table.

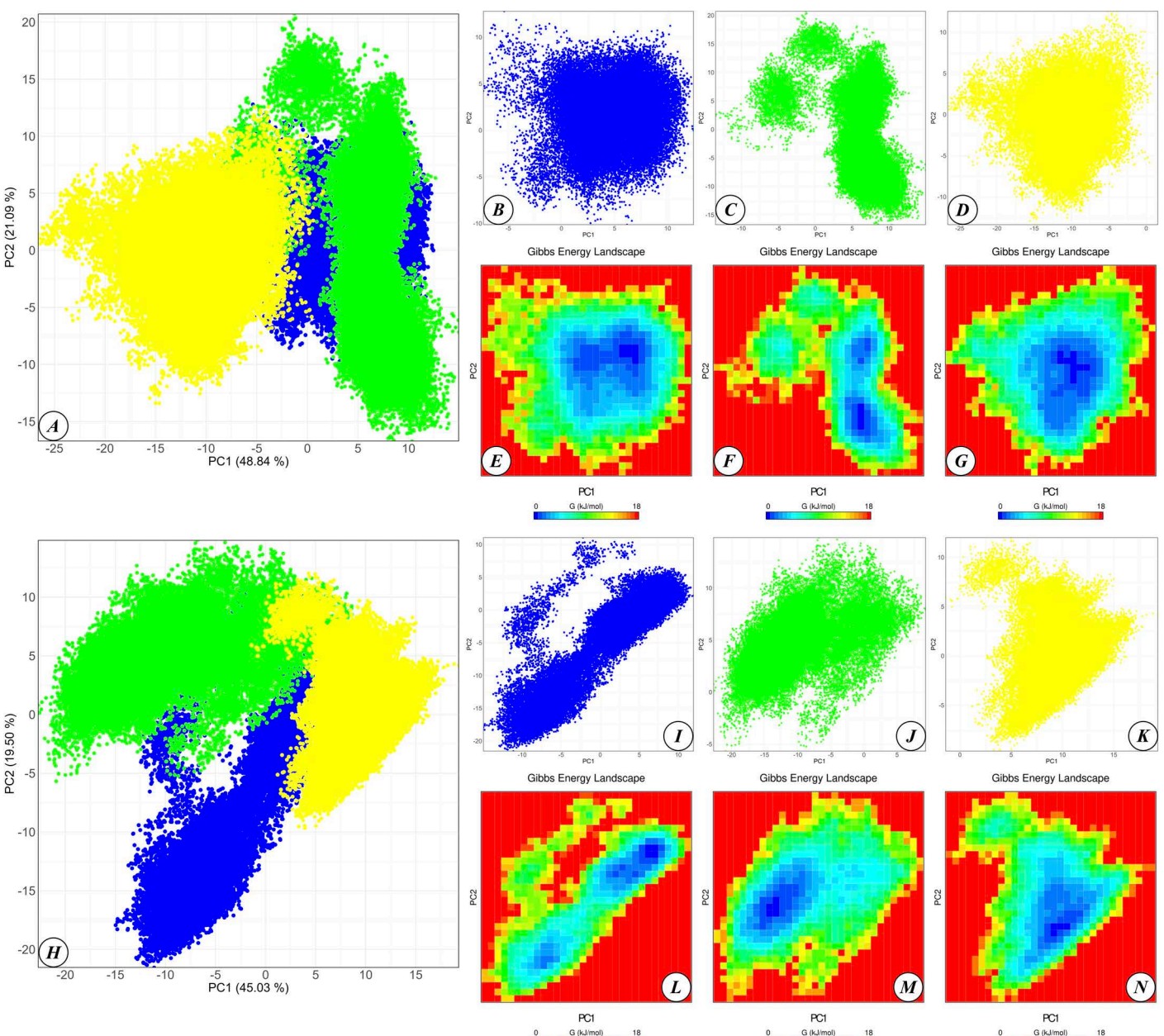

**Fig 7. Principal component analysis (PCA) and Free energy landscape (FEL) analysis. (A)** PCA plot for complexes involving pAPN and RBDs of all three TGEV strains (TGEV Purdue, TGEV 133, and TGEV 133 with additional mutations). **(B–D)** PCA plots for complexes involving pAPN and RBDs of TGEV Purdue **(B)**, TGEV 133 **(C)**, and TGEV 133 with additional mutations **(D)**. **(E–G)** FEL plots for complexes involving pAPN and RBDs of TGEV Purdue **(E)**, TGEV 133 **(F)**, and TGEV 133 with additional mutations **(G)**. **(H)** PCA plot for complexes involving hAPN and RBDs of all three strains. **(I–K)** PCA plots for complexes involving hAPN and RBDs of TGEV Purdue **(I)**, TGEV 133 **(J)**, and TGEV 133 with additional mutations **(K)**. **(L–N)** FEL plots for complexes involving hAPN and RBDs of TGEV Purdue **(L)**, TGEV 133 **(M)**, and TGEV 133 with additional mutations **(N)**. Dots corresponding to complexes involving the Purdue RBD are shown in blue on the PCA plots, those involving the TGEV 133 RBD in green, and those involving the TGEV 133 RBD with additional mutations in yellow. On the FEL plots, conformations with the lowest Gibbs energy are marked in blue, while those with the highest Gibbs energy are marked in red.

indicate that the conformational states of each of the three complexes with pAPN show distinct variability along the first two principal components, reflecting differences in the molecular dynamics of these complexes.

A deeper understanding of the conformational stability of these complexes is achieved through FEL analysis (Fig 7E–G). The FEL plots identify the most stable conformations of each complex, which correspond to regions of lower Gibbs free energy (indicated by blue regions on the FEL plots). Since the FEL plots share the same principal component coordinate system as the PCA plots, the regions with minimal Gibbs free energy for complexes containing the RBDs of TGEV Purdue and TGEV 133 are found to partially overlap. However, the coordinates corresponding to the points of minimal Gibbs free energy for the complex containing the RBD of TGEV 133 with additional mutations are distinct from those of the other two complexes. This observation indicates that the most characteristic stable conformation of the complex including RBD of TGEV 133 with additional mutations differs significantly from the conformations most stable for the TGEV Purdue and TGEV 133 complexes with pAPN.

In similar plots for PCA performed on the molecular dynamics trajectories of C-alpha atoms of complexes comprising hAPN with RBDs of three different virus strains (Fig 7H–K), the first two principal components account for 45.03% and 19.50% of the structural variations observed during molecular dynamics. Some overlap is present in the regions corresponding to complexes including RBDs of TGEV Purdue, TGEV 133, and TGEV 133 with mutations. However, the FEL plots (Fig 7L–N) indicate that the regions of lowest Gibbs free energy are distinct for each of the three complexes, suggesting that these complexes exhibit different stable conformations during molecular dynamics simulations.

In summary, the analysis of various parameters derived from the molecular dynamics trajectories highlights distinct structural features among the complexes containing RBDs of TGEV Purdue, TGEV 133, and TGEV 133 with additional theoretical mutations. Differences in structural dynamics are observed in the interactions of RBDs from different TGEV strains with both pAPN and hAPN. Additionally, the introduction of specific mutations, compared to the reference Purdue strain, can enhance the virus's affinity for the receptor, as indicated by lower $\Delta G_{total}$ values.

## Discussion

The study aimed to investigate the interactions between the RBD of the S-protein in different TGEV and PRCV strains with APN receptors in humans and pigs. The S-protein plays a crucial role in the infectious process of TGEV and PRCV, as evidenced by the deletion distinguishing PRCV from TGEV, which alters the virus tropism and shifts the infection from gastrointestinal to respiratory [22]. This study sought to determine how the virus's infectious potential changes due to single mutations in the RBD and assess the possibility of interspecies transmission from pigs to humans.

The phylogenetic tree (Fig 1) depicts TGEV and PRCV strains isolated in previous years, including recent strains like PRCV USA/ISU20–92330/2020 [28] and TGEV-JMS [61]. This tree is consistent with similar phylogenies reported in previous studies [29,62,63]. Although clustering was based on the amino acid sequences of the S-protein, the phylogenetic relationships align closely with those derived from whole-genome sequences, highlighting the S-protein's value as an informative phylogenetic marker. When multiple strains shared identical S-protein sequences, only one strain was included in the phylogenetic tree, following data from the Identical Protein Groups (NCBI). However, these strains may have variations in other proteins or nucleotide sequences, which may warrant further investigation in studies with other research purposes.

The phylogenetic tree reveals two primary groups. The first group includes traditional TGEV strains [28,29] and most PRCV strains, subdivided into clusters around the Miller M6 and Purdue (PUR46-MAD) strains. The second group comprises evolutionarily younger TGEV strains from the USA, referred to as variant strains [28,29], and the most recently isolated PRCV strains, including USA/ISU20–92330/2020 [28], USA/Minnesota-46140/2016 [29], and OH7269 [64]. Recombination is believed to have occurred between variant TGEV strains and PRCV Minnesota-46140/2016, as their genotypes share unique deletions and amino acid changes [29].

The TGEV 96–1933 strain is phylogenetically distinct from other strains and does not belong to the two main groups. This strain has accumulated significant mutations in the S-protein, particularly in the RBD [65]. Several strains in Fig 1

differ in their overall S-protein sequences but share identical RBD regions. For instance, the RBDs of strains JS2012, TS, ZH, and HN2002 are identical to that of the virulent Miller M6 strain. Consequently, ΔG calculations for RBD-APN complexes involving Miller M6 can be extended to these strains. The JS2012 strain, likely a recombinant of Miller M6 (virulent) and Purdue P115 (attenuated), is highly pathogenic *in vivo*, causing 100% mortality in newborn piglets [66].

Comparison of the S-protein sequences in the RBD region across different coronavirus species revealed that CCoV, which, like TGEV and PRCV, belongs to the *Alphacoronavirus 1* species [67,68], exhibits a sequence closely related to those of TGEV and PRCV strains. Another member of the *Alphacoronavirus* genus, HCoV-229E [69], shares only a limited number of conserved motifs. In contrast, members of the *Betacoronavirus* genus (SARS-CoV, SARS-CoV-2, and MERS-CoV) and the *Deltacoronavirus* genus (PDCoV) display substantial divergence in amino acid sequences within the RBD region. An important aspect of this divergence relates to the receptor usage among these viruses. SARS-CoV and SARS-CoV-2 utilize angiotensin-converting enzyme 2 (ACE2) as their cellular receptor [70], while MERS-CoV binds to dipeptidyl peptidase 4 (DPP4) [71]. In contrast, PDCoV, HCoV-229E, and CCoV all employ APN as a receptor [67–69,72]. Notably, PDCoV, a virus associated with acute enteric disease in pigs, shares receptor usage and tissue tropism with TGEV despite exhibiting a distinct RBD sequence [73]. HCoV-229E provides a compelling example of a coronavirus capable of utilizing human APN, suggesting that adaptation to this receptor type in humans has occurred evolutionarily. Consequently, the possibility of similar adaptations arising in other coronaviruses cannot be excluded.

The study analyzed the ΔG of RBD-APN complexes. Some strains, such as TGEV TO14, TGEV 133, PRCV 135, and PRCV OH7269, form more stable complexes with porcine and human receptors compared to the reference Purdue strain. Conversely, strains like TGEV USA/Minnesota138/2006, PRCV ISU-1, and PRCV 30 exhibit reduced receptor affinity. Notably, strains such as Miller M6 and AHHF demonstrate ΔG values comparable to the Purdue strain (–57.78 kcal/mol, –57.68 kcal/mol, and –57.83 kcal/mol with pAPN, respectively). The Purdue strain is attenuated [74,75], while Miller M6 and AHHF are virulent [62,74]. Miller M6 and AHHF differ from Purdue by four and six mutations in the RBD, respectively, alongside additional mutations in other regions of the S-protein and other viral proteins. This suggests that attenuation may not necessarily result from reduced RBD-APN affinity. Previous studies have addressed how specific mutations in TGEV correlate with virulence [74]. Given that the AHHF strain carries mutations previously associated with attenuation, this remains a debatable issue, meriting further investigation.

The ΔG calculations indicate that the most stable RBD-APN complexes are formed with the TGEV 133 strain. Specifically, the ΔG values for the RBD of this strain in complexes with porcine and human APN receptors are –63.51 kcal/mol and –60.68 kcal/mol, respectively. Notably, the complex between the RBD of TGEV 133 and hAPN is more stable than those formed by some strains with pAPN, including the reference Purdue strain. This finding suggests the potential for the virus to penetrate human cells. These results also apply to several TGEV strains isolated in Korea, such as 133, KT2, and KT3, which share identical RBD amino acid sequences. These strains form a distinct phylogenetic branch (Fig 1) and exhibit high genetic similarity [76]. The increased affinity of the TGEV 133 strain for APN receptors is attributed to three mutations compared to the reference Purdue strain. While the H562N (H560N) substitution is common among many strains, the N563K (N561K) and D600H (D598H) mutations are specific to TGEV 133 and closely related strains (Fig 2, S2 Table). This example demonstrates how single, random mutations can significantly alter the virus's receptor affinity.

To explore whether further stabilization of RBD-APN complexes is possible, the study analyzed additional theoretical single mutations introduced into the TGEV 133 RBD. These theoretical mutations, arising from random missense single nucleotide substitutions, were found to enhance complex stability. For instance, the P547T (P545T), T551I (T549I), and T551K (T549K) mutations each had a significant stabilizing effect. A double mutation (P547T+T551K) produced even greater stabilization, with ΔG values for complexes with pAPN and hAPN reaching –70.96 kcal/mol and –67.53 kcal/mol, respectively. Remarkably, the RBD-hAPN complex in this case demonstrated greater stability than any RBD-pAPN complex formed by native TGEV or PRCV strain.

Molecular dynamics analysis further confirmed differences in the structural properties of RBD-APN complexes formed by the Purdue strain, the TGEV 133 strain, and mutated TGEV 133. Parameters such as RMSD, RMSF, radius of gyration, SASA, and the number of hydrogen bonds and non-polar contacts between the RBD and receptor revealed structural variations. Principal component analysis (PCA) and free energy landscape (FEL) analysis demonstrated differences in predominant conformational states among these complexes. Binding free energy analysis of the molecular dynamics trajectories confirmed that the TGEV 133 strain has higher affinity for both human and porcine APN receptors compared to the Purdue strain. Moreover, molecular dynamics simulations confirmed that additional mutations in strain 133 can increase RBD-hAPN affinity to levels observed in interactions between TGEV RBDs and pAPN.

Although this study is based on computational approaches, it provides valuable insights and highlights promising directions for future research. These include *in vitro* testing of interactions between TGEV and PRCV strains, or their individual RBDs, with cells displaying potential host receptors, as well as systematic monitoring of emerging mutations in newly identified isolates of these viruses. Moreover, any scenario in which the affinity of TGEV or PRCV RBDs for human receptors approaches or surpasses their affinity for porcine receptors warrants careful consideration regarding the potential for interspecies transmission. This underscores the theoretical possibility that evolutionary processes could lead to mutations in the RBD of the S-protein, increasing the affinity of TGEV or PRCV strains for human receptors. Furthermore, mutations that enhance affinity for human receptors simultaneously increase affinity for porcine receptors, suggesting that such mutations could accumulate within pig populations before potentially spreading to humans through contact. The risks of this scenario are supported by recent detections of two canine coronavirus strains, CCoV-HuPn-2018 and HuCCoV_Z19, in human samples [77,78]. These strains also belong to *Alphacoronavirus 1*, like TGEV and PRCV, and utilize APN for receptor binding [79]. Given the findings of this study and previous research, continued surveillance of mutations in TGEV and PRCV, as well as related *Alphacoronavirus 1* strains infecting other animal species [67], remains critical for assessing the potential risks of interspecies transmission and the emergence of new coronavirus epidemics.

## Conclusions

This study employed *in silico* methods to analyze the TGEV and PRCV strains, which are responsible for diseases in pigs, with the aim of predicting their potential for interspecies transmission to humans. The sequences of their S-proteins were examined to assess phylogenetic relationships among virus strains and to evaluate the conservation of sequences in their RBD regions. The interactions between the RBDs of 16 TGEV and PRCV strains and their cellular receptor, APN, were studied, and the binding free energies of the corresponding complexes were calculated. The analysis revealed that the RBDs of these coronaviruses exhibit significantly higher affinity for porcine APN than for human APN ($p < 0.05$). However, it was shown that the viruses' affinity for the human receptor could theoretically be increased through mutations. Specifically, the double mutation P547T+T551K in the TGEV 133 strain, which stabilizes the RBD-APN complex, was found to be sufficient to bring the binding free energy of the complex with the human receptor to a level comparable to virus interaction with the porcine receptor. Given the critical role of the binding process in the infection mechanism of coronaviruses, it is essential to consider the potential risks that TGEV and PRCV may pose to humans if mutations accumulate and the viruses continue to evolve.

## Supporting information

**S1 Table. Spike protein sequences of TGEV and PRCV strains from NCBI.**
(XLSX)

**S2 Table. Differences in amino acids of receptor-binding domains of TGEV and PRCV strains. Grantham's distances for the corresponding (by multiple alignment) pairs of amino acids are given in the last column.** Distances

0–50 are considered as conservative substitutions (green), 51–100 as moderately conservative (yellow), 101–150 as moderately radical (orange), or ≥151 as radical (red).
(XLSX)

**S3 Table. Prediction on the ΔΔG (kcal/mol) of the complexes, involving pig APN and receptor-binding domain of the TGEV 133 with single mutations.** Cells with negative and neutral ΔΔG values are colored in green, mutations with a mostly stabilizing effect are displayed in yellow.
(XLSX)

**S4 Table. Prediction on the ΔΔG (kcal/mol) of the complexes, involving human APN and receptor-binding domain of the TGEV 133 with single mutations.** Cells with negative and neutral ΔΔG values are colored in green, mutations with a mostly stabilizing effect are displayed in yellow.
(XLSX)

**S5 Fig. Contacts at the interface of RBD-APN complexes.** (A–C) Interactions between pig APN and RBDs of the Purdue strain (A), 133 strain (B), and 133 strain with additional mutations (C). (D–F) Interactions between human APN and RBDs of the Purdue strain (D), 133 strain (E), and 133 strain with additional mutations (F). Chain A corresponds to APN receptors, and chain B to RBDs. H-bonds are represented as blue lines, while non-bonded contacts are depicted as orange striped lines, with stripes' width proportional to the number of atomic contacts. Positive residues are colored blue, negative residues red, neutral residues green, aliphatic residues gray, aromatic residues purple, and proline and glycine residues orange.
(TIF)

**S6 Table. RMSD, RMSF, Rg, SASA, H-bonds and non-polar contacts calculated from molecular dynamics trajectories.**
(XLSX)

**S7 Table. Binding free energy components calculated from molecular dynamics trajectories.**
(XLSX)

## Acknowledgments

The authors are grateful to Lubov Atramentova (for providing advice on statistical analysis).

## Author contributions

**Conceptualization:** Mykyta Peka, Viktor Balatsky.

**Data curation:** Mykyta Peka.

**Formal analysis:** Mykyta Peka, Viktor Balatsky.

**Funding acquisition:** Mykyta Peka, Viktor Balatsky.

**Investigation:** Mykyta Peka, Viktor Balatsky.

**Methodology:** Mykyta Peka.

**Project administration:** Viktor Balatsky.

**Resources:** Mykyta Peka.

**Supervision:** Viktor Balatsky.

**Validation:** Viktor Balatsky.

**Visualization:** Mykyta Peka.

**Writing – original draft:** Mykyta Peka, Viktor Balatsky.

**Writing – review & editing:** Mykyta Peka, Viktor Balatsky.

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
