## [Editor Report · Decision Letter 0]

18 Feb 2025

PONE-D-25-05046

Binding of transmissible gastroenteritis virus and porcine respiratory coronavirus to human and porcine aminopeptidase N receptors as an indicator of cross-species transmission

PLOS ONE

Dear Dr. Peka,

Thank you for submitting your manuscript to PLOS ONE. After careful consideration, we have decided that your manuscript does not meet our criteria for publication and must therefore be rejected.

Specifically, this study used only in silico prediction, which is not sufficient to go to conclusions. The biochemical or virological experiments are technically required to validate the results.

I am sorry that we cannot be more positive on this occasion, but hope that you appreciate the reasons for this decision.

Kind regards,

Li Xing

Academic Editor

PLOS ONE

- - - - -

---

## [Author Response · Author response to Decision Letter 1]

28 Feb 2025

Dear Editor,

We appreciate the time and effort the Editorial team have dedicated to reviewing our manuscript, “Binding of transmissible gastroenteritis virus and porcine respiratory coronavirus to human and porcine aminopeptidase N receptors as an indicator of cross-species transmission” (PONE-D-25-05046), submitted to PLOS ONE. We respect your decision; however, we would like to formally appeal the rejection and kindly request a reconsideration of our manuscript. We believe that our study aligns with the journal’s scope and publication criteria, and we would like to clarify specific aspects that may have contributed to the initial decision.

The primary reason for rejection, as stated in the decision letter, is that our study relies on in silico methods and does not include biochemical or virological experiments. However, we would like to highlight that PLOS ONE has previously published purely in silico studies in the fields of virology and computational biology. Our work follows a rigorous computational methodology, including phylogenetic analysis, molecular modeling, binding free energy calculations, and molecular dynamics simulations, that are established techniques in computational biology. These approaches have been widely used in reputable published literature to study host-virus interactions and assess infectious potential of viruses.

We believe that our study meets PLOS ONE’s criteria for publication, as it provides novel insights into the potential for interspecies transmission of coronaviruses and adheres to established methodological standards in computational biology. Before submitting our manuscript, we carefully reviewed PLOS ONE’s policies, publication criteria, and submission guidelines, as well as previously published studies in the journal. Based on this review, we concluded that our study falls within the journal’s scope and that our manuscript complies with all formatting and data reporting requirements.

We acknowledge that there may have been some misunderstandings regarding the study’s objectives and the implications of our computational findings. To address this, we have revised some sections of our manuscript to more clearly emphasize that our study focuses on assessing coronavirus-receptor interactions using computational methods and that any conclusions drawn about potential interactions remain within the scope of our results. We have noted the corresponding changes in the “Track Changes” version of our manuscript. We fully acknowledge that experimental validation would further strengthen the findings, and we have carefully framed our conclusions within the scope of in silico predictions, avoiding overstatement of biological implications. We believe that this study provides a valuable framework that can guide future laboratory research in the field.

We kindly request a reevaluation of our manuscript, either by reconsidering its suitability for peer review or by providing further clarification on the journal’s policy regarding in silico research. If additional revisions or clarifications would make the study more appropriate for consideration, we would be happy to make those adjustments.

We sincerely appreciate your time and consideration and look forward to your response.

Best regards,

Authors

---

## [Decision Letter · Decision Letter 1]

28 Mar 2025

PONE-D-25-05046R1Binding of transmissible gastroenteritis virus and porcine respiratory coronavirus to human and porcine aminopeptidase N receptors as an indicator of cross-species transmissionPLOS ONE

Dear Dr. Peka,

Thank you for submitting your manuscript to PLOS ONE. After careful consideration, we feel that it has merit but does not fully meet PLOS ONE’s publication criteria as it currently stands. Therefore, we invite you to submit a revised version of the manuscript that addresses the points raised during the review process.

We look forward to receiving your revised manuscript.

Kind regards,

Sheikh Arslan Sehgal, PhD

Academic Editor

PLOS ONE

Journal Requirements:

1. Please ensure that your manuscript meets PLOS ONE's style requirements, including those for file naming. The PLOS ONE style templates can be found at https://journals.plos.org/plosone/s/file?id=wjVg/PLOSOne_formatting_sample_main_body.pdf and https://journals.plos.org/plosone/s/file?id=ba62/PLOSOne_formatting_sample_title_authors_affiliatios.pdf

“This study was funded by the National Academy of Agrarian Sciences of Ukraine (grant registration number: 0124U002088).”

3. Please amend your Response to Reviewers letter to include a point by point response to each of the points made by the Editor and / or Reviewers. Please follow this link for more information: http://blogs.PLOS.org/everyone/2011/05/10/how-to-submit-your-revised-manuscript/

Additional Editor Comments (if provided):

Reviewers' comments:

Reviewer's Responses to Questions

**Comments to the Author**

1. If the authors have adequately addressed your comments raised in a previous round of review and you feel that this manuscript is now acceptable for publication, you may indicate that here to bypass the “Comments to the Author” section, enter your conflict of interest statement in the “Confidential to Editor” section, and submit your "Accept" recommendation.

Reviewer #1: (No Response)

Reviewer #2: All comments have been addressed

2. Is the manuscript technically sound, and do the data support the conclusions?

Reviewer #1: (No Response)

Reviewer #2: Yes

3. Has the statistical analysis been performed appropriately and rigorously? 

Reviewer #1: (No Response)

Reviewer #2: I Don't Know

4. Have the authors made all data underlying the findings in their manuscript fully available?

Reviewer #1: (No Response)

Reviewer #2: Yes

5. Is the manuscript presented in an intelligible fashion and written in standard English?

Reviewer #1: (No Response)

Reviewer #2: Yes

6. Review Comments to the Author

Reviewer #1: here is my comments on "Binding of transmissible gastroenteritis virus and porcine .." Manu.

the manuscript needs improvements in deiffernt aspects; as follows

(1) While the authors have effectively used Grantham’s Distance (Table 1) to assess the conservatism or radicality of amino acid substitutions, incorporating multiple sequence alignment (MSA) could further enhance the analysis by providing insights into the evolutionary conservation of these residues across different species. This would help in determining whether the observed substitutions occur in highly conserved regions, which could indicate functional significance.

(2) The authors have utilised the Hawdock server for their binding free energy calculations. However, an alternative approach such as gmx_MMPBSA, which integrates molecular dynamics simulations with MM/PBSA calculations in GROMACS, could provide a more dynamic and detailed estimation of ΔG. Could the authors clarify why they opted for Hawdock instead of employing gmx_MMPBSA for free energy calculations?"

(3) The manuscript would benefit from additional graphical representations of molecular interactions. More detailed visualizations of key interactions, such as hydrogen bonds, hydrophobic contacts, and salt bridges, could enhance the clarity of the findings.

(4) Regarding the calculated ΔΔG values for the mutations, could the authors clarify whether these calculations were performed using the PDB structures before or after molecular dynamics (MD) simulations? Using post-MD structures could provide a more realistic representation of the mutated protein’s stability and interactions.

(5) The authors have calculated ΔG in the gas phase. Could they explain why this approach was used, considering that molecular interactions typically occur in aqueous environments?

Reviewer #2: This manuscript shows novel aspects of TGEV and PRCV at the possibility of zoonotic pathogens.

The following points should be considered for the clearness of this manuscript.

line 198 to 199 : on the on the HawkDock...  on the HawkDock...

7. PLOS authors have the option to publish the peer review history of their article (what does this mean? ). If published, this will include your full peer review and any attached files.

**Do you want your identity to be public for this peer review?** For information about this choice, including consent withdrawal, please see our Privacy Policy .

Reviewer #1: No

Reviewer #2: **Yes: ** Yun Sang Cho

---

## [Author Response · Author response to Decision Letter 2]

5 Apr 2025

Response to Editor:

The authors thank the PLOS ONE editorial board for reviewing the submitted manuscript. The authors have carefully considered the comments and suggestions provided by the editors and reviewers and have undertaken the necessary revisions to ensure that the manuscript meets the high quality standards and the PLOS ONE’s publication criteria.

1. The manuscript has been checked to ensure full compliance with PLOS ONE’s style requirements.

2. We would like to state that the funder had no role in study design, data collection and analysis, decision to publish, or preparation of the manuscript. As there was no option to do this in Editorial Manager while revising the manuscript, we ask to add an additional sentence to our financial disclosure statement and present it the following form:

“This study was funded by the National Academy of Agrarian Sciences of Ukraine (grant registration number: 0124U002088). The funder had no role in study design, data collection and analysis, decision to publish, or preparation of the manuscript.”

3. A point-by-point response has been provided to address each of the reviewers’ comments, and the necessary revisions of the manuscript have been made.

4. The authors consent to the publication of the peer review history of this manuscript in accordance with PLOS ONE’s publication policies (https://journals.plos.org/plosone/s/editorial-and-peer-review-process#loc-peer-review-history).

The authors remain fully committed to collaborating with the PLOS ONE editorial board to ensure that the manuscript meets the necessary quality standards.

Response to Reviewer 1:

The authors thank the Reviewer for evaluation of this manuscript, suggestions, and comments. We have addressed the following points raised in the review to enhance the clarity and overall quality of the manuscript:

Reviewer #1: here is my comments on "Binding of transmissible gastroenteritis virus and porcine .." Manu.the manuscript needs improvements in deiffernt aspects; as follows

(1) While the authors have effectively used Grantham’s Distance (Table 1) to assess the conservatism or radicality of amino acid substitutions, incorporating multiple sequence alignment (MSA) could further enhance the analysis by providing insights into the evolutionary conservation of these residues across different species. This would help in determining whether the observed substitutions occur in highly conserved regions, which could indicate functional significance.

Authors’ response. The authors fully agree with the Reviewer’s observation that multiple alignment can enhance the depth of the analysis. In response, an additional multiple alignment was performed, incorporating spike protein sequences from other Coronaviridae family members that are relevant to the study of human and animal infections. These include canine coronavirus (CCoV) and HCoV-229E, both alphacoronaviruses that utilize the APN receptor, similar to TGEV and PRCV, as well as SARS-CoV, SARS-CoV-2, and MERS-CoV, which are betacoronaviruses responsible for human epidemics and utilize different receptors. Additionally, porcine deltacoronavirus (PDCoV), an emerging porcine enteropathogenic virus that, despite being distinct from TGEV, also utilizes APN, was included. The results of this alignment are presented in Fig 2.

Conducting this multiple alignment not only provided additional valuable insights into the interspecies conservation of the spike protein at the RBD but also allowed for the optimization of result presentation. Fig 2 now more clearly illustrates the positions of amino acid substitutions both between TGEV and PRCV strains and at the interspecies level. Furthermore, by incorporating a color panel based on Grantham’s distance categorization (conservative, moderately conservative, moderately radical, and radical substitutions), the conservation of these substitutions is effectively highlighted.

Given these enhancements, the authors concluded that Fig 2 presents the findings more comprehensively than the previously included table and therefore deserves placement in the main part of the manuscript. The table containing Grantham’s distance estimates for substitutions at specific positions, which was originally presented in this context, has been converted into an editable tabular format and relocated to the Supporting data files (S2 Table), ensuring it remains accessible for interested readers.

The authors express their sincere gratitude to the Reviewer for this insightful suggestion. Implementing multiple alignment not only led to additional significant findings and a deeper discussion but also improved the clarity and structure of results presentation.

(2) The authors have utilised the Hawdock server for their binding free energy calculations. However, an alternative approach such as gmx_MMPBSA, which integrates molecular dynamics simulations with MM/PBSA calculations in GROMACS, could provide a more dynamic and detailed estimation of ΔG. Could the authors clarify why they opted for Hawdock instead of employing gmx_MMPBSA for free energy calculations?"

Authors’ response. The authors would like to highlight that both the Hawkdock server and the gmx_MMPBSA approach were used in this study for binding free energy calculations. The Hawkdock server was employed for the initial analysis and comparison of different TGEV and PRCV strains, as well as for screening theoretical mutations that could enhance the affinity of the RBD to pig and human APN-receptors. The binding free energy calculations performed using the Hawkdock server are based on the MM/GBSA approach, which is quite accurate. Additionally, the Hawkdock server allows for a balance between computational efficiency and accuracy, which was important for the comparative analysis of different strains and the screening of the set of potential mutations.

Furthermore, the gmx_MMPBSA approach was applied to assess the binding free energy of RBD-APN complexes selected for molecular dynamics simulations. As correctly noted by the Reviewer, this method enabled a more detailed estimation of ΔG in a dynamic context, accounting for potential changes in ΔG associated with conformational changes during molecular dynamics simulations.

The authors have made every effort to clearly specify in the Materials and Methods section the contexts in which each approach (Hawkdock server or gmx_MMPBSA) was used for ΔG calculations.

(3) The manuscript would benefit from additional graphical representations of molecular interactions. More detailed visualizations of key interactions, such as hydrogen bonds, hydrophobic contacts, and salt bridges, could enhance the clarity of the findings.

Authors’ response. In accordance with the recommendations, the authors have expanded the graphical presentation. In addition to the general view of the RBD-APN complex (Fig 3A), the manuscript now includes images illustrating the interface interaction of the virus’s RBD with pAPN (Fig 3B) and hAPN (Fig 3C). Furthermore, the authors would like to notice that a detailed schematic representation of the various interface contacts in the RBD-APN complexes is available in supplementary figure (S5 Figure).

(4) Regarding the calculated ΔΔG values for the mutations, could the authors clarify whether these calculations were performed using the PDB structures before or after molecular dynamics (MD) simulations? Using post-MD structures could provide a more realistic representation of the mutated protein’s stability and interactions.

Authors’ response. In our study, ΔΔG calculations for single amino acid mutations were performed using reference structures based on the conformation and orientation of the RBD and APN in a previously defined crystallographic RBD-APN complex (PDB: 4F5C). This approach ensured structurally reliable starting points for mutation screening and maintained consistency in the calculations. ΔΔG prediction tools are designed to operate with static structures rather than molecular dynamics (MD)-derived ensembles, making this a common approach in stability prediction studies.

Following MD simulations of selected complexes, instead of recalculating ΔΔG, we computed the ΔG values for these complexes throughout the MD simulations. This approach provides comparative insights into the stability of complexes over time, effectively capturing the same information as post-MD ΔΔG calculations. Since ΔΔG is defined as the difference between two ΔG values (ΔΔG = ΔG2 - ΔG1), tracking ΔG values across MD also allowed us to assess the relative stability of complexes dynamically.

(5) The authors have calculated ΔG in the gas phase. Could they explain why this approach was used, considering that molecular interactions typically occur in aqueous environments?

Authors’ response. The authors agree with the Reviewer that molecular interactions occur in aqueous environments and would like to emphasize that solvation effects were accounted for in binding free energy calculations. Both the Hawkdock server and gmx_MMPBSA tool incorporate solvation effects in their methodologies. Specifically, the Hawkdock server accounts for polar and non-polar solvation contributions when calculating ΔG.

For ΔG values derived using gmx_MMPBSA tool based on molecular dynamics simulations, the authors have reported several energy components in the figure titled “Analysis of virus binding to APN through molecular dynamics.” This includes molecular mechanics energy in the gas phase (Figure 5G-H), solvation energy contributions (Figure 5I-J), and total binding free energy (Figure 5E-F). Additionally, in the current version of the manuscript, Table 8 has been added to provide a detailed breakdown of each energy component, including molecular mechanics (bonded interactions, van der Waals forces, electrostatic and non-bonded interactions), as well as polar and non-polar solvation energy contributions to the total binding free energy.

Thus, the binding free energy calculations account for both molecular mechanics energy in the gas phase and solvation effects.

The authors sincerely thank the Reviewer once again for all valuable recommendations, which have significantly improved the quality of this manuscript. We remain fully committed to making any refinements if necessary to ensure that the manuscript meets the high quality standards.

Response to Reviewer 2:

Reviewer #2: This manuscript shows novel aspects of TGEV and PRCV at the possibility of zoonotic pathogens.

The following points should be considered for the clearness of this manuscript.

line 198 to 199 : on the on the HawkDock...  on the HawkDock...

Authors’ response. The authors are grateful to the Reviewer for evaluation of the manuscript. The authors made adjustments and removed duplication in the line:

…on the on the HawkDock...  on the HawkDock...

We would like to emphasize that, during the revision process, the manuscript was thoroughly proofread to ensure that it is well-written, with particular attention to language accuracy and clarity of presentation. We remain fully committed to making any refinements if necessary to ensure that the manuscript meets the high quality standards.

---

## [Decision Letter · Decision Letter 2]

20 Apr 2025

PONE-D-25-05046R2Binding of transmissible gastroenteritis virus and porcine respiratory coronavirus to human and porcine aminopeptidase N receptors as an indicator of cross-species transmissionPLOS ONE

Dear Dr. Peka,

Thank you for submitting your manuscript to PLOS ONE. After careful consideration, we feel that it has merit but does not fully meet PLOS ONE’s publication criteria as it currently stands. Therefore, we invite you to submit a revised version of the manuscript that addresses the points raised during the review process.

We look forward to receiving your revised manuscript.

Kind regards,

Sheikh Arslan Sehgal, PhD

Academic Editor

PLOS ONE

Journal Requirements:

Reviewers' comments:

Reviewer's Responses to Questions

**Comments to the Author**

1. If the authors have adequately addressed your comments raised in a previous round of review and you feel that this manuscript is now acceptable for publication, you may indicate that here to bypass the “Comments to the Author” section, enter your conflict of interest statement in the “Confidential to Editor” section, and submit your "Accept" recommendation.

Reviewer #1: (No Response)

2. Is the manuscript technically sound, and do the data support the conclusions?

Reviewer #1: (No Response)

3. Has the statistical analysis been performed appropriately and rigorously? 

Reviewer #1: (No Response)

4. Have the authors made all data underlying the findings in their manuscript fully available?

Reviewer #1: (No Response)

5. Is the manuscript presented in an intelligible fashion and written in standard English?

Reviewer #1: No

6. Review Comments to the Author

Reviewer #1: the manuscript is improved, while the images of simulation study regarding the plots and so on should be reformat for publication, and also it would be better to represent the interactions by contribution of forces.

7. PLOS authors have the option to publish the peer review history of their article (what does this mean? ). If published, this will include your full peer review and any attached files.

**Do you want your identity to be public for this peer review?** For information about this choice, including consent withdrawal, please see our Privacy Policy .

Reviewer #1: No

---

## [Author Response · Author response to Decision Letter 3]

23 Apr 2025

Response to Editor:

The authors thank the PLOS ONE Editorial Board for their time and effort in reviewing the submitted manuscript. We appreciate the valuable comments provided by the Editor and Reviewers, which have contributed significantly to enhancing the quality of the manuscript.

In accordance with the Editor’s instructions, the reference list was carefully reviewed to ensure its accuracy. We also confirm that it does not include citations of any retracted sources.

In response to the Reviewer’s suggestion regarding figure formatting for publication, we would like to note that all figures have been processed using the Preflight Analysis and Conversion Engine (PACE) digital diagnostic tool, as recommended. This step ensures that the figures fully comply with PLOS formatting requirements.

The authors reaffirm their commitment to continued collaboration with the Editorial Board and Reviewers to maintain the high quality of the manuscript and ensure its full adherence to PLOS ONE’s publication standards.

Response to Reviewer 1:

The authors thank the Reviewer for the time dedicated to re-evaluating the revised manuscript. In response to the Reviewer’s comments, the images included in the manuscript were carefully reviewed to ensure that all titles and labels are free of grammatical errors, and that the images fully comply with the PLOS ONE submission guidelines, including those related to format, resolution, dimensions, etc.

Following the Reviewer’s recommendation to represent the interactions by the contribution of forces, the authors have added Figure 6 (numbered accordingly in the current version of the manuscript). This figure illustrates the dynamics of multiple interaction components, including van der Waals interaction energy, electrostatic interaction energy, and both polar and non-polar solvation energy components. These additions enhance the informational value and presentation of the results.

The authors are grateful for the Reviewer’s constructive suggestions, which have helped improve the overall quality of the manuscript. We remain committed to continued collaboration with the Editorial Board and Reviewer throughout the editorial process.

---

## [Decision Letter · Decision Letter 3]

5 May 2025

Binding of transmissible gastroenteritis virus and porcine respiratory coronavirus to human and porcine aminopeptidase N receptors as an indicator of cross-species transmission

PONE-D-25-05046R3

Dear Dr. Peka,

We’re pleased to inform you that your manuscript has been judged scientifically suitable for publication and will be formally accepted for publication once it meets all outstanding technical requirements.

Kind regards,

Sheikh Arslan Sehgal, PhD

Academic Editor

PLOS ONE

Additional Editor Comments (optional):

Reviewers' comments:

Reviewer's Responses to Questions

**Comments to the Author**

1. If the authors have adequately addressed your comments raised in a previous round of review and you feel that this manuscript is now acceptable for publication, you may indicate that here to bypass the “Comments to the Author” section, enter your conflict of interest statement in the “Confidential to Editor” section, and submit your "Accept" recommendation.

Reviewer #1: All comments have been addressed

2. Is the manuscript technically sound, and do the data support the conclusions?

Reviewer #1: Partly

3. Has the statistical analysis been performed appropriately and rigorously? 

Reviewer #1: Yes

4. Have the authors made all data underlying the findings in their manuscript fully available?

Reviewer #1: Yes

5. Is the manuscript presented in an intelligible fashion and written in standard English?

Reviewer #1: Yes

6. Review Comments to the Author

Reviewer #1: after revisions now the manuscript is improved and therefor it might be suitable for consideration.

7. PLOS authors have the option to publish the peer review history of their article (what does this mean? ). If published, this will include your full peer review and any attached files.

**Do you want your identity to be public for this peer review?** For information about this choice, including consent withdrawal, please see our Privacy Policy .

Reviewer #1: No

---

## [Editor Report · Acceptance letter]

PONE-D-25-05046R3

PLOS ONE

Dear Dr. Peka,

I'm pleased to inform you that your manuscript has been deemed suitable for publication in PLOS ONE. Congratulations! Your manuscript is now being handed over to our production team.

Kind regards,

on behalf of

Dr Sheikh Arslan Sehgal

Academic Editor

PLOS ONE